# Amino acids stimulate the endosome-to-Golgi trafficking through Ragulator and small GTPase Arl5

Meng Shi[1], Bing Chen[1], Divyanshu Mahajan[1], Boon Kim Boh[1], Yan Zhou[1], Bamaprasad Dutta[1], Hieng Chiong Tie[1], Siu Kwan Sze [1], Geng Wu[2] & Lei Lu [1]

The endosome-to-Golgi or endocytic retrograde trafficking pathway is an important post-Golgi recycling route. Here we show that amino acids (AAs) can stimulate the retrograde trafficking and regulate the cell surface localization of certain Golgi membrane proteins. By testing components of the AA-stimulated mTORC1 signaling pathway, we demonstrate that SLC38A9, v-ATPase and Ragulator, but not Rag GTPases and mTORC1, are essential for the AA-stimulated trafficking. Arl5, an ARF-like family small GTPase, interacts with Ragulator in an AA-regulated manner and both Arl5 and its effector, the Golgi-associated retrograde protein complex (GARP), are required for the AA-stimulated trafficking. We have therefore identified a mechanistic connection between the nutrient signaling and the retrograde trafficking pathway, whereby SLC38A9 and v-ATPase sense AA-sufficiency and Ragulator might function as a guanine nucleotide exchange factor to activate Arl5, which, together with GARP, a tethering factor, probably facilitates the endosome-to-Golgi trafficking.

[1] School of Biological Sciences, Nanyang Technological University, 60 Nanyang Drive, Singapore 637551, Singapore. [2] State Key Laboratory of Microbial Metabolism, School of Life Sciences and Biotechnology, Shanghai Jiao Tong University, Shanghai 200240, China. These authors contributed equally: Meng Shi, Bing Chen. Correspondence and requests for materials should be addressed to L.L. (email: lulei@ntu.edu.sg)

In eukaryotic cells, proteins and lipids (cargos) are dynamically exchanged among cellular organelles through trafficking routes or pathways. In the endocytic pathway, cargos on the plasma membrane (PM) are internalized to the early endosome (EE). From the EE, cargos can be degraded in the lysosome via the later endosome (LE). Alternatively, they can take the retrograde or the endosome-to-Golgi trafficking pathway to the *trans*-Golgi network (TGN), where they return to either the PM or the endosome to complete their itinerary cycles[1–6]. A growing list of cargos, including most TGN resident transmembrane proteins (TGN membrane proteins), has been documented to take the retrograde route. Pathogens, such as viruses and plant or bacterial toxins, can also hijack this pathway to invade cells while avoiding lysosomal degradation. As a major cellular recycling pathway, the endosome-to-Golgi trafficking has been recognized for its roles in the post-Golgi secretion, biogenesis of the lysosome, maintenance of sphingolipid homeostasis, regulation of Wnt signaling, and pathogenesis of neurodegenerative diseases[7–9].

Recent progress in this field offers us a rough picture on how the endosome-to-Golgi trafficking works at molecular and cellular level[1,3–5,10]. First, cargos are sorted into a membrane carrier at the endosomal membrane in conjunction with coats, coat adaptors, retromer, Golgi-associated retrograde protein complex (GARP), and actin cytoskeleton. Next, the budded carrier is targeted to the TGN by microtubule motors. It then attaches to the TGN membrane by tethering factors such as Golgins and GARP. Finally, the formation of SNARE complex drives the fusion between the carrier and TGN, accomplishing the delivery of cargos.

Nutrient, including amino acids (AAs), glucose, and other carbon sources, is the most fundamental resource for the growth and proliferation of cells. Nutrient sufficiency stimulates anabolic metabolism, such as the biosynthesis of macromolecules and the biogenesis of organelles; on the other hand, nutrient starvation triggers catabolic pathways, such as autophagy, to break down macromolecules in order to recycle much needed materials for cell survival. Cells must have evolved sophisticated signaling networks to coordinate their sub-cellular activities according to the environment nutrient. For example, in yeast, AA starvation causes the accumulation of Gap1p on the PM to scavenge extracellular nitrogen sources; whereas the presence of AAs, especially Gln, activates TORC1 signaling cascade to internalize and degrade Gap1p[11–14]. Atg9, a conserved transmembrane protein essential for autophagy, translocates from the periphery to phagophore assembly site in yeast[15] or from the Golgi to endosome via Ulk1-dependent pathway in mammalian cells[16]. Besides Atg9, it is currently unknown in mammalian cells if and how nutrient regulates intracellular membrane trafficking, especially the endosome-to-Golgi pathway.

In contrast, a great deal of molecular details have been known on how AAs regulate cellular metabolism through transcription and translation. The cell's metabolic decision is mainly made through the mechanistic target of rapamycin complex 1 (mTORC1) signaling pathway, which senses the presence of nutrient and growth factors in combination with the cellular energy and stress status[17–19]. AA sufficiency first triggers SLC38A9[20–22], a SLC-family AA transceptor, and v-ATPase[23], a proton pump responsible for the acidification of the lysosome. Next, activated SLC38A9 and v-ATPase signal to Ragulator by rearranging their interaction with the latter. Ragulator is a pentameric complex comprising Lamtor1-5[24,25]. Following the activation, Ragulator functions as the guanine nucleotide exchange factor (GEF) for heterodimeric Rag GTPases[24]. Finally, GTP-loaded Rag heterodimer recruits mTORC1 to the lysosomal surface[25], where the full kinase activity of mTORC1 is turned on by growth-factor-activated small GTPase, Rheb[26]. Active mTORC1 initiates anabolic processes through translation and transcription by phosphorylating a cascade of its substrates.

Here, we ask if nutrient can regulate the endosome-to-Golgi trafficking and demonstrate that the trafficking is promoted by AAs. Our study uncovers a mechanistic connection between the AA-sensing module of the mTORC1 signaling pathway and the endosome-to-Golgi trafficking machinery components including Arl5 and GARP.

## Results

**Starvation translocates TGN membrane proteins to endosomes.** To investigate if nutrient plays a role in the endocytic membrane trafficking, we compared the sub-cellular distribution of TGN membrane proteins in normal and starvation medium. Most TGN membrane proteins, such as furin, TGN46, cation-independent mannose 6-phosphate receptor (CI-M6PR), cation-dependent mannose 6-phosphate receptor (CD-M6PR), and sortilin, cycle between the PM and TGN through endosomes[1]. Their relative distribution between the Golgi and endosomal pool shifts in response to a change in the endocytic trafficking. In the complete medium (DMEM supplemented with 10% fetal bovine serum), endogenous furin mainly colocalized with Golgin-245, as expected (Fig. 1a, b). When serum or growth factor was withdrawn by incubation in DMEM for 1 h, no significant change of furin was observed (Fig. 1a, b). In contrast, when cells were starved of both AAs and growth factors by incubating in Hank's balanced salt solution (HBSS) for 1 h, furin lost its Golgi pool and appeared diffused throughout the cytosol (Fig. 1a, b). The starvation-induced translocation was reversible. When HBSS-treated cells were subsequently supplied with nutrient (DMEM), furin rapidly re-appeared at the Golgi (Fig. 1c). The finding was also observed using exogenously expressed full-length furin-GFP (Fig. 1d and Supplementary Fig. 1a). The rapid and reversible distribution demonstrated that furin was probably not degraded and, instead, it was likely arrested at the peripheral pool. Similar but less striking effect was also observed for TGN46 (Supplementary Fig. 1b).

To further confirm our observation and resolve the peripheral location of furin, we imaged exogenously expressed CD8a-furin chimera, which consists of CD8a luminal and transmembrane domain and furin cytosolic domain[27], and made similar observation (Fig. 1e, f). Under starvation, the peripheral pool of CD8a-furin colocalized with the EE marker, RUFY1[28], and the recycling endosome (RE) marker, transferrin receptor (TfR-GFP)[29], but not the LE marker, GFP-Rab7[30] (Fig. 1g), suggesting that it mainly localized to the EE and RE during starvation. The colocalization study using full length furin confirmed these results, although a significant localization to the LE was also observed (Supplementary Fig. 1c), probably due to the contribution of native transmembrane domain[31]. The preferential endosomal distribution of CD8a-furin under starvation was also biochemically demonstrated using sucrose gradient ultracentrifugation (Fig. 1h, i), in which endosomal and TGN fractions were identified by EEA1 and syntaxin6, respectively[32,33].

Starvation-induced change of localization was also similarly observed for other TGN membrane proteins such as endogenous CI-M6PR and overexpressed CD8a-CI-M6PR (Supplementary Fig. 1d-g). Our subsequent studies mainly focused on CD8a-furin since its subcellular localization seems to be most sensitive to nutrient among TGN membrane reporters that we tested.

**AAs stimulate the retrograde trafficking.** The reduction of the Golgi pool and the concomitant increase in the endosomal pool under HBSS treatment suggest that DMEM and complete medium might stimulate the endosome-to-Golgi trafficking. The complete medium comprises serum, which is the source of

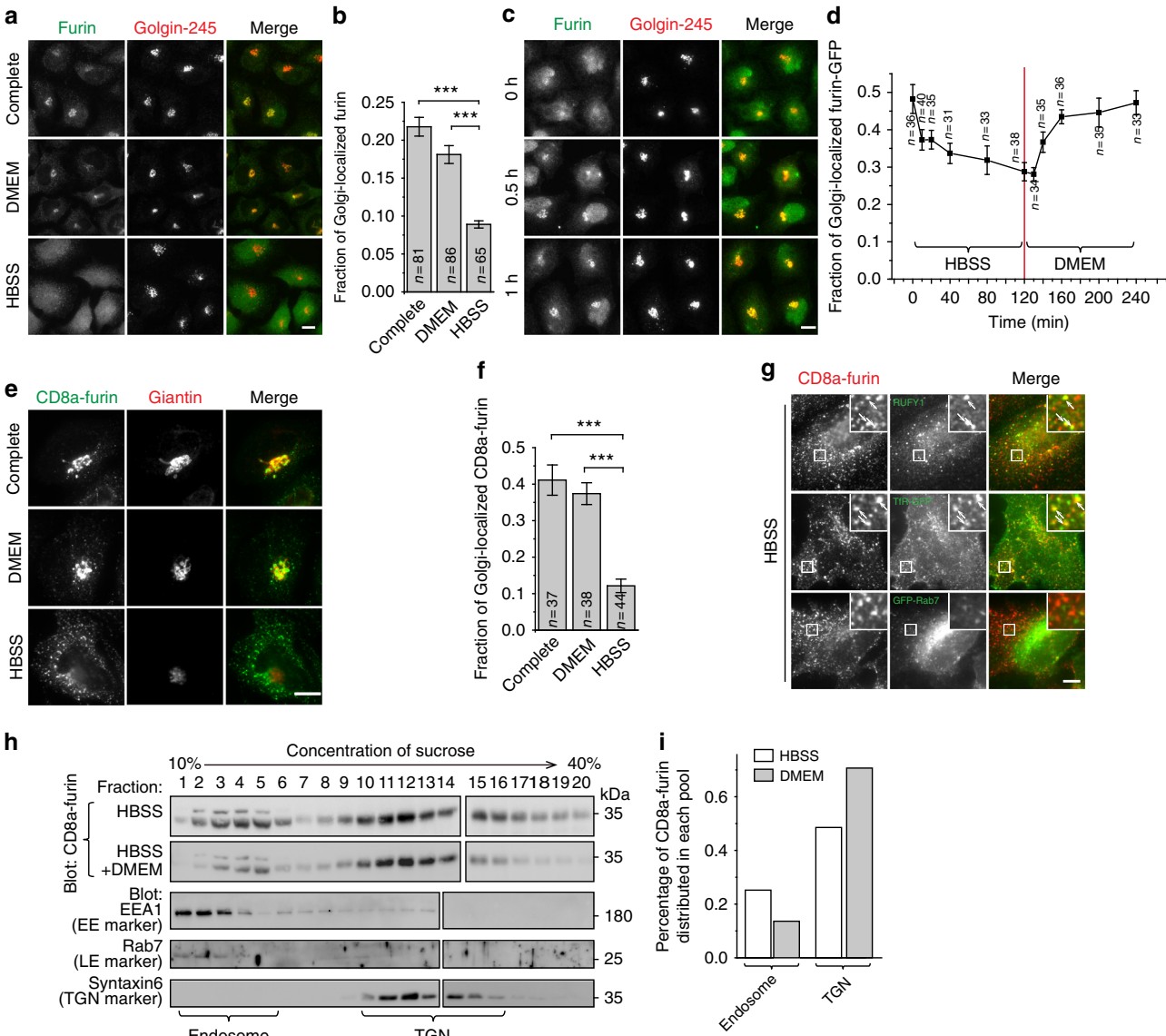

**Fig. 1** Starvation translocates TGN membrane proteins to endosomes. All cells are HeLa cells. **a**, **b** Furin loses its Golgi localization during starvation. Cells treated with indicated medium for 1 h and endogenous furin and Golgin-245 were stained. The fraction of Golgi-localized furin is quantified in **b**. **c** The recovery of Golgi localization of furin after supplying nutrient. After starvation in HBSS for 2 h, cells were treated with DMEM for indicated time and stained as in **a**. **d** Kinetics of Golgi-localized furin-GFP during HBSS and subsequent DMEM treatment. Cells expressing furin-GFP were first starved in HBSS for 2 h and subsequently stimulated by DMEM for 2 h. At indicated time, cells were stained for endogenous Giantin and the fraction of Golgi-localized furin-GFP is quantified. **e**, **f** Nutrient starvation significantly reduces the Golgi localization of CD8a-furin. Cells transiently expressing CD8a-furin were treated by indicated medium for 2 h and stained as in **e**. The fraction of Golgi-localized CD8a-furin is quantified in **f**. **g** The translocation of CD8a-furin to the endosome during nutrient starvation. Cells transiently expressing indicated constructs were treated with HBSS for 2 h and stained. Boxed regions are enlarged at the upper right corner. Arrows indicate colocalization. **h**, **i** The endosomal pool of CD8a-furin increases during nutrient starvation. Similar results have been observed in four independent experiments. Cells stably expressing CD8a-furin were treated with HBSS for 2 h or with additional 20 min treatment of DMEM. Lysates were subjected to sucrose gradient centrifugation to separate organelles. 20 fractions were collected and immunoblotted for CD8a-furin and markers. Percentages of CD8a-furin distributed in the endosomal (fractions 1–5) and TGN pool (fractions 10–16) are quantified in **i**. **b**, **d**, and **f** are representative results from three independent experiments. Complete, complete medium; n, the number of cells analyzed; error bar, mean ± s.e. m.; scale bar, 10 μm. P values were from t test (unpaired and two-tailed). ***$P \leq 0.0005$

growth factors, and DMEM, which consists of DMEM-base (inorganic salts and vitamins), AAs (15 AAs including Gln), and glucose. To reveal the component(s) behind the stimulation, the PM-to-Golgi trafficking assay was conducted in the testing medium comprising DMEM-base supplemented with combinations of dialyzed serum, AAs and glucose. To that end, CD8a-furin-expressing cells were first starved in DMEM-base. The surface-exposed CD8a-furin was labeled and its trafficking to the Golgi was subsequently monitored. We found that AAs, but not glucose or serum, were sufficient to stimulate the endocytic trafficking of CD8a-furin to the Golgi (Fig. 2a, b). Supplementation of AAs, dialyzed serum and glucose in DMEM-base, or the usage of the complete medium, produced no more stimulatory effect than AAs alone (Fig. 2a, b). In fact, a weaker stimulation by the complete medium than AAs was often observed, suggesting an unknown adverse effect of glucose and growth factors on the endocytic trafficking. Similar AA-stimulation effect was also observed in BSC-1 cells or by using other reporters, such

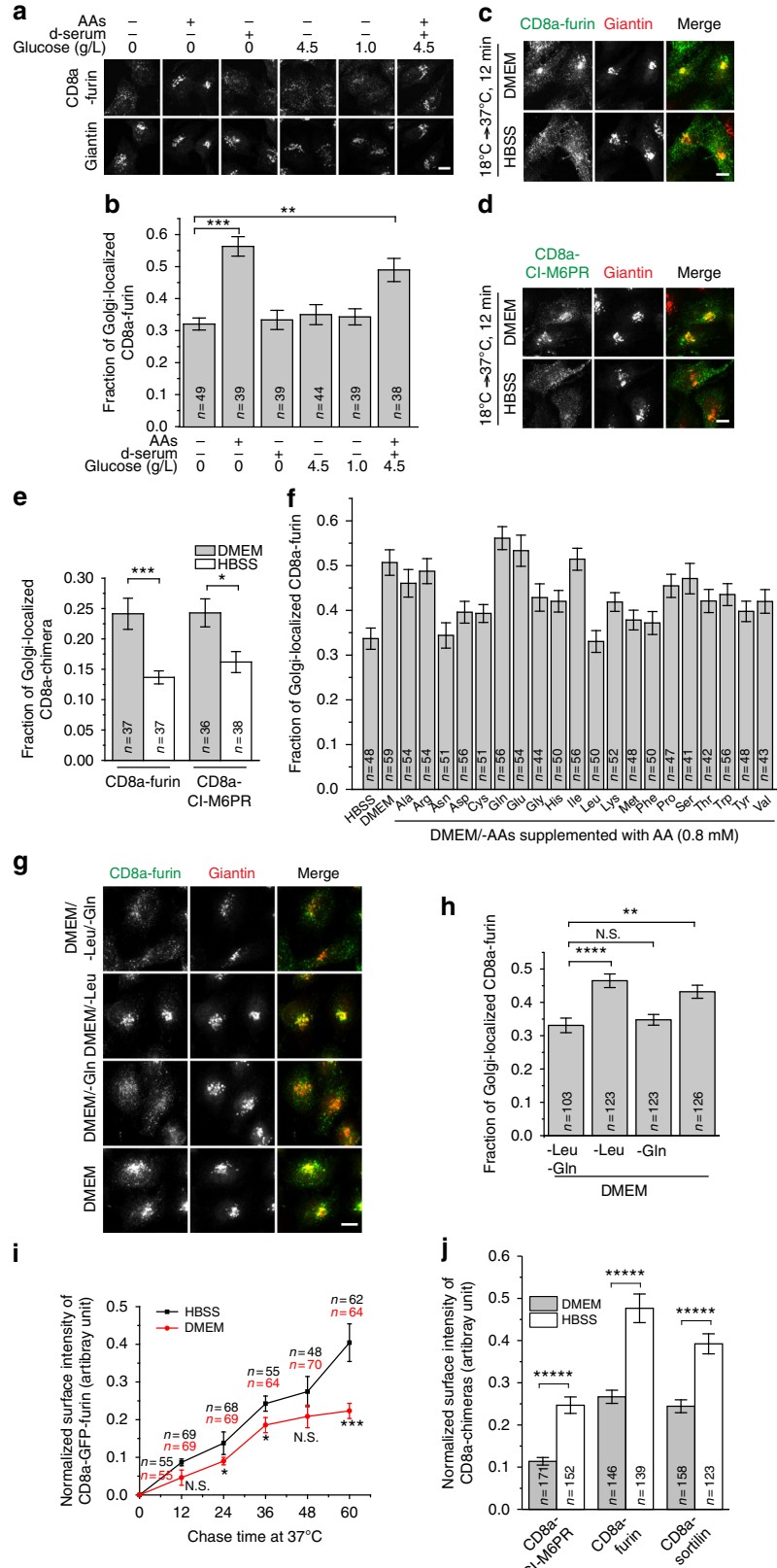

as CD8a-fused CI-M6PR, CD-M6PR and sortilin (Supplementary Fig. 2a, b) and Shiga toxin B fragment (STxB) (Supplementary Fig. 2c, d); however, the Golgi trafficking of Cholera toxin B fragment was not significantly affected by AAs (Supplementary Fig. 2e, f).

The endocytic trafficking of CD8a-furin to the Golgi comprises two consecutive steps, the clathrin-dependent endocytosis to the endosome and the subsequent endosome-to-Golgi trafficking. Our quantitative analysis indicated that the internalized CD8a-furin within 6 min of chase did not display an obvious difference

**Fig. 2** AAs stimulate the endosome-to-Golgi trafficking. All cells are HeLa cells. **a**, **b** AAs but not growth factors and glucose stimulate the retrograde trafficking to the Golgi. Cells stably expressing CD8a-furin were treated with DMEM-base for 2 h. The surface-exposed CD8a-furin was labeled by anti-CD8a antibody and chased in respective medium for 20 min. The fraction of CD8a-furin at the Golgi is quantified after staining. d-serum, 10% dialyzed serum. **c-e** AAs stimulate the endosome-to-Golgi trafficking. Cells transiently expressing CD8a-furin or CD8a-CI-M6PR were surface labeled by anti-CD8a antibody and synchronized at endosomes at 18 °C in HBSS for 2 h before being chased at 37 °C in HBSS or DMEM for 12 min. The fraction of Golgi-localized CD8a-chimeras is quantified after staining. **f-h** Gln has one of the most acute stimulating effects on endocytic trafficking to the Golgi. Similar to **a**, **b** except that the nutrient starvation was conducted in HBSS before surface labeling. The labeled CD8a-furin was then chased in HBSS, DMEM, DMEM/-AAs supplemented with indicated AA at 0.8 mM in **f** or DMEM selectively leaving out indicated AA(s) in **g**. The fraction of Golgi-localized CD8a-furin is quantified in **h**. **i** AAs decrease the Golgi-to-PM trafficking of furin. Cells expressing SBP-GFP-CD8a-furin were treated with biotin for 2 h at 20 °C before the system was warmed up during the chase. The surface-labeling intensity was normalized by the total cellular GFP intensity. **j** AAs reduce cell surface CD8a-furin, -sortilin, and -CI-M6PR. Transfected cells were labeled for both surface and intracellular pools of CD8a-chimeras. The intensity of the surface pool was normalized by that of the intracellular pool. **b**, **e**, **f**, **h-j** are representative results from three independent experiments. Scale bar, 10 μm; *n*, the number of cells analyzed; error bar, mean ± s.e.m.; *P* values were from *t* test (unpaired and two-tailed). *N.S.* not significant (*P* > 0.05); *\*P* ≤ 0.05; *\*\*P* ≤ 0.005; *\*\*\*P* ≤ 0.0005; *\*\*\*\*P* ≤ 0.00005; *\*\*\*\*\*P* ≤ 0.000005

between AA sufficiency and starvation (Supplementary Fig. 2g, h), therefore suggesting that the endocytosis could not be the target of AA stimulation. Hence, we directly tested the retrograde trafficking. Antibody-labeled CD8a-furin or -CI-M6PR was first allowed to accumulate at the EE and RE using 18 °C synchronization protocol[34,35]. The Golgi localization was subsequently quantified after a chase at 37 °C for 12 min in the medium with or without AAs. We found that AAs significantly increased the Golgi-localized CD8a-furin and -CI-M6PR (Fig. 2c–e). In summary, our data demonstrate that AAs stimulate the endosome-to-Golgi trafficking of TGN membrane proteins. Upon AA starvation, the endosome-to-Golgi trafficking is compromised; as the Golgi-to-PM exocytic trafficking proceeds, an elevated endosomal pool is resulted for the TGN membrane protein at the expense of its Golgi one.

**Gln stimulates the retrograde trafficking most acutely**. To determine which AA(s) is(are) responsible for the AA-stimulated endosome-to-Golgi trafficking, we tested each of 20 AAs at the same concentration (0.8 mM). While a majority of AAs stimulated the trafficking to various degrees, Gln, a non-essential AA, consistently stood out as the most potent stimulator (Fig. 2f). Glu also displayed a strong effect, probably due to its intracellular conversion to Gln by glutamine synthetase. The stimulating effect of Gln was concentration dependent and peaked at ~0.6 mM (Supplementary Fig. 2i). Other stimulatory AAs, such as Ala, followed similar trend. To our surprise, Leu, which is an essential AA and the strongest stimulator for mTORC1 signaling[36], as well as Asn, displayed minimal stimulation (Fig. 2f). The lack of stimulation by Leu was probably not due to suboptimal concentration used in our assay since similar result was observed for a wide range of concentrations from 0.01 to 5.12 mM (Supplementary Fig. 2i). We found that the effect of combining different AAs appeared complex and was not simply additive (Supplementary Fig. 2j), the mechanism behind which awaits future exploration. The observation is nonetheless consistent with the finding that DMEM, which contains 15 AAs, repeatedly showed less activity than Gln alone (Fig. 2f). It is possible that AAs can positively or negatively modulate each other's activities, considering that certain AAs can act as exchangers for others[37].

To further test if Gln is essential for the retrograde trafficking in parallel comparison with Leu, we investigated the subcellular distribution of CD8a-furin in DMEM selectively leaving out Gln (DMEM/-Gln), Leu (DMEM/-Leu), or both (DMEM/-Leu/-Gln) (Fig. 2g, h). In media without Gln (DMEM/-Leu/-Gln and DMEM/-Gln), CD8a-furin mainly localized to peripheral puncta at the expense of its Golgi localization, indicating an essential role of Gln in the retrograde trafficking; in media containing Gln

(DMEM/-Leu and DMEM), CD8a-furin prominently accumulated in the Golgi. In contrast, the depletion of Leu (DMEM/-Leu) did not affect the Golgi localization of CD8a-furin comparing with DMEM. We ruled out the possibility that our HeLa cells became insensitive to Leu since, consistent with our current knowledge[38–41], Gln was observed to be essential and synergize with Leu for the activation of mTORC1 signaling (Supplementary Fig. 2k). Collectively, we conclude that Gln, but not Leu, is a necessary and sufficient stimulator for the endosome-to-Golgi trafficking.

**AAs regulate the surface presence of Golgi membrane proteins**. We next asked if the exocytic trafficking of furin, including the Golgi-to-PM and endosome-to-PM pathways, are also sensitive to AAs. First, we measured the cell surface arrival kinetics of a synchronized furin reporter from the Golgi (see Methods). Interestingly, it was found that HBSS stimulated the Golgi-to-PM trafficking of furin reporter in comparison to DMEM (Fig. 2i). For furin, its endocytosis is insensitive to AAs (see above) and its endosome-to-PM recycling was reported to be negligible[42]. Since the cell surface furin is consumed by the endocytosis and contributed by the exocytosis, our finding suggests a possible decrease of furin at the cell surface upon AA stimulation, which was demonstrated by surface labeling using CD8a-furin (Fig. 2j). Similar trend was also noted for sortilin and CI-M6PR using their CD8a-chimeras (Fig. 2j).

To test if other surface proteins can be regulated by AAs, we performed an unbiased screen of surface-biotinylated proteins under HBSS or DMEM treatment using stable isotope labeling with amino acids in cell culture (SILAC) based mass spectrometry. Among ~430 proteins that were identified with statistical significance (*P* ≤ 0.05) (Supplementary Fig. 2l, m), 85 were transmembrane, GPI-anchored or secretory proteins according to UniProt annotations (Supplementary Table 1). The remaining hits were likely false positives or pulled down indirectly. We focused on seven Golgi membrane proteins that displayed significant changes (Table 1). The cell surface presence of SorLA and VIP36 was found reduced upon AA stimulation. SorLA, a sortilin-related protein, was the only TGN resident and its DMEM/HBSS-ratio is the lowest. The rest five are all Golgi type-II transmembrane proteins, including GPP130, GP73, and three Golgi enzymes, and their cell surface localizations are upregulated by AA stimulation. While GPP130 and GP73 were reported to cycle between the PM and Golgi[43] and the surface presence of Golgi glycosyltransferases was acknowledged long ago[44], it is unclear how these proteins, which all have extremely short cytosolic tails, are retrieved from their post-Golgi localization. It is tempting to speculate that Vps10 family receptors, such as

**Table 1 List of Golgi transmembrane proteins with AA-sensitive surface localization. In total, there are seven Golgi transmembrane proteins among candidate hits with $P$ value $\leq 0.05$ and $\log_2$(DMEM/HBSS-ratio) $\geq 0.5$ or $\leq -0.3$**

| UniProt identifier | Protein name | Cell surface DMEM/HBSS-ratio mean ± s.d. |
|---|---|---|
| A0A024R3H2 | SorLA | 0.3 ± 0.4 |
| B4DWN1 | VIP36 | 0.8 ± 0.2 |
| Q59G70 | MGAT1 | 1.5 ± 0.4 |
| B4DLB8 | GalT | 1.9 ± 0.7 |
| Q8NBJ4 | GP73 | 1.9 ± 0.7 |
| O00461 | GPP130 | 3.2 ± 0.9 |
| Q9H1B5 | XylT2 | 4 ± 1 |

SorLA and sortilin, function to retrieve them and therefore clear their presence at the cell surface. While the molecular and cellular mechanism and biological significances of the data require further investigation, we showed that cell surface presentation of Golgi membrane proteins can be regulated by AAs.

**Components regulating the AA-stimulated trafficking**. It is known that nutrient signaling culminates in the activation of mTORC1 through SLC38A9, v-ATPase, Ragulator and hetero-dimeric Rag GTPases[20–25]. To test if AA-stimulated retrograde trafficking utilizes a similar pathway, we selectively compromised each component through small molecule inhibitors or RNAi-mediated knockdowns and subsequently investigated the ensuing effect on the trafficking. To compare and contrast the stimulatory effect, the fraction of Golgi-localized CD8a-furin under AA stimulation was normalized by that under starvation to yield a quantity referred to as the AA-stimulated Golgi trafficking. In the presence of concanamycin A (conA), an inhibitor of v-ATPase[45], the AA-stimulated Golgi trafficking decreased significantly in comparison with the control (Fig. 3a; Supplementary Fig. 3a). When SLC38A9, a high affinity transporter for Gln[21], was depleted (Fig. 3b), the AA-stimulated mTORC1 activity reduced significantly as expected[20–22] (Fig. 3c) and so did AA-stimulated Golgi trafficking (Fig. 3d). Similarly, when Lamtor1 and Lamtor3, two subunits of Ragulator, were individually depleted (Fig. 3e; Supplementary Fig. 3b), the AA-stimulated Golgi trafficking decreased significantly in comparison with the control (Fig. 3f), implying that Ragulator is also required. For Lamtor1, the specificity of our knockdown was demonstrated in the rescue experiment by expressing the RNAi-resistant Lamtor1 (Fig. 3g, h). Knockdown of Lamtor1 also inhibited the AA-stimulated reduction of CD8a-furin at the cell surface, suggesting an essential role of Ragulator-mediated endocytic trafficking in the surface localization of furin (Fig. 3i).

Our findings on the AA-stimulated trafficking so far seem consistent with the AA-stimulated mTORC1 signaling pathway. However, simultaneous depletion of both Rag A and B by shRNAs (Fig. 3e, f), a condition in which AA-stimulated mTORC1 activation was inhibited (Supplementary Fig. 3c), did not significantly affect the AA-stimulated Golgi trafficking (Fig. 3f). When mTORC1 activity was strongly inhibited by rapamycin or Torin1 (Supplementary Fig. 3d), the AA-stimulated Golgi trafficking did not significantly change either (Fig. 3j). Altogether, our results demonstrated that SLC38A9, v-ATPase, and Ragulator, but not Rag GTPases and mTORC1, are probably involved in the AA-stimulated endosome-to-Golgi trafficking.

**Arl5b interacts with Ragulator through Lamtor1**. The endosome-to-Golgi trafficking requires Rab and Arl-family small GTPases[1,3,4,10] and we previously discovered Arl1 as a key regulator for this pathway[46,47]. In our quest for additional Arl regulators for this pathway, we focused on Arl5b, which resides on the Golgi and regulates the endosome-to-Golgi and the reverse trafficking, probably by interacting with its effector GARP and AP4, respectively[48–50]. To gain insight into the upstream regulators and downstream effectors of Arl5b, we performed a yeast two-hybrid screen of human kidney cDNA library using GTP-mutant form of Arl5b as the bait. Interestingly, one of the strongest hits identified was Lamtor1, a subunit of Ragulator.

The interaction between Arl5b and Lamtor1 was first biochemically confirmed in pull-down and immunoprecipitation assays. Bead-immobilized GST-Arl5b and a control Arl, GST-Arl1, were in vitro loaded with guanosine 5′-[β,γ-imido]tripho-sphate (GMPPNP; a non-hydrolyzable GTP analog) and GDP. The resulting beads were then incubated with HEK293T cell lysate expressing Lamtor1-GFP. Both GMPPNP and GDP-loaded Arl5b pulled down Lamtor1-GFP (Fig. 4a). In contrast, neither GMPPNP nor GDP-loaded Arl1 retained Lamtor1-GFP (Fig. 4a), demonstrating the specificity of Arl5b–Lamtor1 interaction. Similar to Arl1 and other Ras-family small GTPases[47], guanine nucleotide binding mutations, Q70L (GTP-mutant form; hereafter QL) and T30N (GDP-mutant form; hereafter TN), were introduced to Arl5b to make it constitutively active and inactive, respectively[49]. Using HEK293T cell lysates co-expressing C-terminally GFP-tagged Arl5b-wild type (hereafter wt), QL, TN, or Lamtor2 together with Lamtor1-Myc or SNX3-Myc, we found that Arl5b-QL and -TN, but not Arl5b-wt, strongly interacted with Lamtor1 in both forward and reverse co-immunoprecipitations (co-IPs) (Fig. 4b, c). As previously reported[24,25], exogenously expressed Lamtor2-GFP interacted with Lamtor1-Myc (Fig. 4b, c). In contrast, corresponding Arl1 mutants did not co-IP Lamtor1 (Supplementary Fig. 4a).

To investigate Arl5b-binding region of Lamtor1, we generated GFP-tagged serial truncates of Lamtor1. When HEK293T cell lysates expressing individual truncates were incubated with bead-immobilized GST-Arl5b-TN, it was found that both N- or C-terminus of Lamtor1 are required for the interaction (Fig. 4d). Previous studies have established that Lamtor1 anchors Ragulator to the lysosomal membrane by its N-terminal dual-lipid modification and it functions as a scaffold to independently bind to two heterodimeric subcomplexes—Lamtor2–3 and Lamtor4–5[24,25,51]. We characterized the interaction between Arl5b and individual subunits or subcomplexes. Except for Lamtor1, immobilized GST-Arl5b-QL or -TN did not pull down individually expressed Lamtor2, 3, 4, and 5 (Fig. 4e). When incubated with cell lysates expressing combinations of exogenously expressed Ragulator subunits, immobilized GST-Arl5b pulled down Lamtor2–3 and Lamtor4–5 subcomplexes only in the presence of co-expressed Lamtor1 (Fig. 4f). In addition to exogenously expressed Lamtors, immobilized GST-Arl5b also pulled down endogenous Lamtors (Fig. 4g). In summary, we conclude that Arl5b interacts with Ragulator through Lamtor1. Although both GTP and GDP-mutant forms interacted with Lamtor1, GDP-mutant form of Arl5b appeared to interact more strongly (Fig. 4b, c, e), the significance of which is discussed later. Ragulator is also known to interact with heterodimeric Rag GTPases via RagA or RagB loaded with GDP[24]. We observed that an excess amount of GST-Arl5b-TN, but not GST, significantly reduced the amount of RagB-T54L (GDP-mutant form) and RagC IPed by Lamtor1 (Fig. 4h), suggesting that Arl5b and Rag might interact with Ragulator in a mutually exclusive manner.

In human and mouse genome, there are three paralogs of Arl5, Arl5a, b, and c, with AA sequence identity ≥64%. In contrast to mouse Arl5c, human Arl5c is significantly different from the rest paralogs as it does not have a typical G3 box (Supplementary

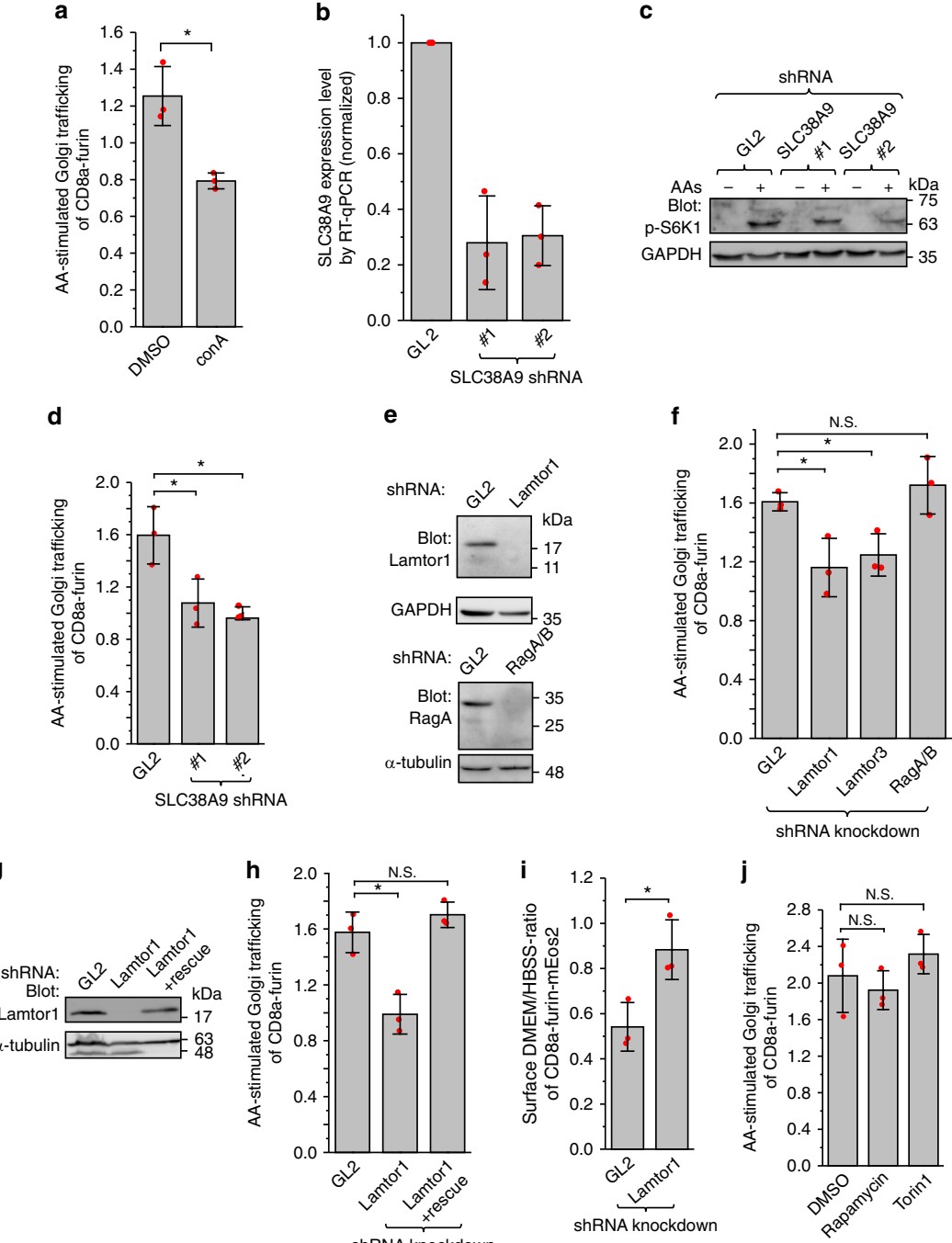

**Fig. 3** Signaling components essential for the AA-stimulated retrograde trafficking. All cells are HeLa cells. **a** Cells stably expressing CD8a-furin were starved in HBSS for 2 h followed by surface-labeling and subsequent incubation with either HBSS or DMEM for 20 min. 1% DMSO or 2.5 μM conA was present throughout the incubation. Cells were stained and the AA-stimulated Golgi trafficking is quantified by imaging. **b** Endogenous SLC38A9 was depleted by lentivirus-transduced shRNAs as assessed by RT-qPCR from $n = 3$ independent experiments. **c** The knockdown of endogenous SLC38A9 attenuated the AA-stimulated mTORC1 activity. Knockdown cells were incubated with DMEM/-AAs for 50 min followed by incubation with DMEM for 20 min. Cell lysates were immuno-blotted for phospho-S6K1 (p-S6K1) and GAPDH. **d** SLC38A9 is required for the AA-simulated Golgi trafficking. Cells treated with indicated shRNAs were transfected to express CD8a-furin and subjected to treatment and analysis similar to **a**. **e** Immuno-blots showing that endogenous Lamtor1 and RagA/B were depleted by respective lentivirus-transduced shRNAs. **f** Lamtor1 and 3 but not RagA/B are required for the AA-stimulated Golgi trafficking. The experiment was similar to **d**. **g, h** Under Lamtor1 knockdown condition similar to **e**, an RNAi-resistant Lamtor1 was able to express and rescue the AA-stimulated Golgi trafficking. **i** Lamtor1 is required for the AA-stimulated reduction of surface CD8a-furin-mEos2. Knockdown and surface labeling were similar to **e** and Fig. 2j, respectively. Surface intensity was normalized by mEos2 total intensity. Surface DMEM/HBSS-ratio is the normalized surface intensity under DMEM divided by that under HBSS treatment. **j** mTORC1 is not required for the AA-stimulated retrograde trafficking. The experiment was conducted similarly to **a** except that 1% DMSO, 100 nM rapamycin, or 250 nM Torin1 was present throughout the treatment. In **a, b, d, f, h–j**, the displayed value is the mean of $n = 3$ independent experiments and individual data points are shown as red dots. Error bar, mean ± s.d.; $P$ values were from $t$ test (unpaired and two-tailed); *N.S.*, not significant ($P > 0.05$); *$P \leq 0.05$. GL2 is a non-targeting control siRNA or shRNA

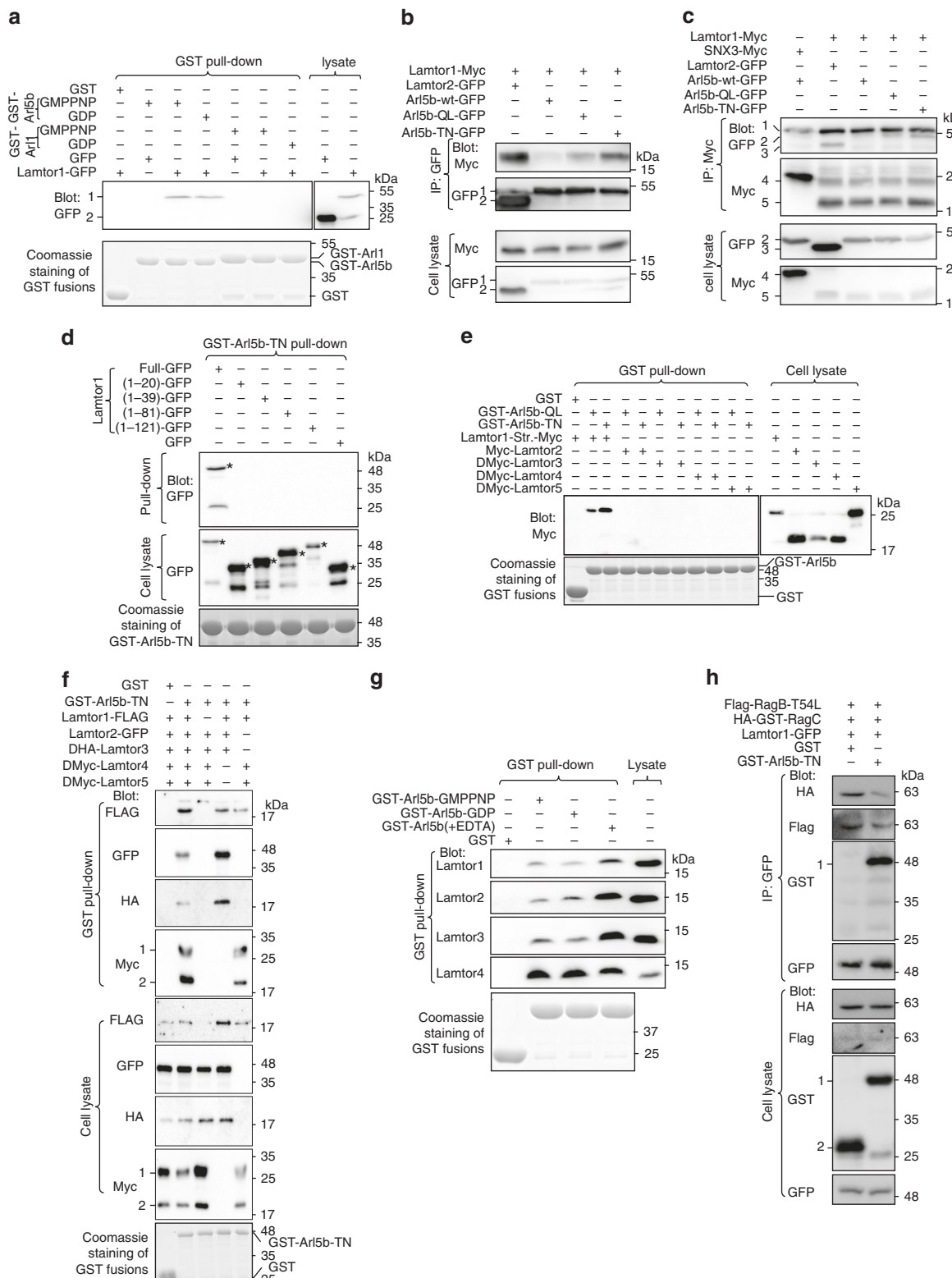

Fig. 4b). Using purified cDNA plasmids as calibration standards, our reverse transcription quantitative PCR (RT-qPCR) revealed that transcripts of Arl5a and b are roughly equal while that of Arl5c is ~30 folds less than Arl5a and b in HeLa cells, consistent with the proposal that *Arl5c* gene could be a pseudogene[48]. In agreement with their high sequence identity, immobilized GST-Arl5a, Arl5b, and mouse Arl5c pulled down Lamtor1-GFP,

though Arl5b appeared to retain the most (Supplementary Fig. 4c). These results suggest that Arl5a, Arl5b, and mouse Arl5c could have redundant cellular functions but Arl5b possibly contributes most to Ragulator interaction.

**Arl5b colocalizes with Lamtor1 at the endolysosome.** When transiently expressed in HeLa cells, C-terminally GFP-tagged wt,

**Fig. 4** Arl5b interacts with Ragulator through Lamtor1. HEK293T cells were used. **a** Arl5b, but not Arl1, specifically pulled down Lamtor1-GFP. Bead-immobilized GST-Arl5b or Arl1 was in vitro loaded with GDP or GMPPNP and subsequently incubated with cell lysates expressing GFP or Lamtor1-GFP. Pull-downs were analyzed by immuno-blotting GFP fusions. The loading of GST fusions were shown by Coomassie blue staining. 1 and 2 indicate Lamtor1-GFP and GFP band, respectively. **b, c** Arl5b-QL and -TN interact with Lamtor1-Myc in forward and reverse co-IPs. Cells co-expressing indicated tagged-proteins were incubated with indicated antibodies and IPs were immuno-blotted against indicated tags. Lamtor2-GFP and SNX3-Myc served as a positive and negative control, respectively. In **b**, 1 and 2 indicate Arl5b-(wt, QL or TN)-GFP and Lamtor2-GFP band, respectively. In **c**, 1–5 indicate IgG heavy chain, Arl5b-(wt, QL or TN)-GFP, Lamtor2-GFP, SNX3-Myc, and Lamtor1-Myc band, respectively. **d** Full length Lamtor1 is required for Arl5b–Lamtor1 interaction. Bead-immobilized GST-Arl5b-TN was incubated with cell lysates expressing indicated fragments of Lamtor1, and pull-downs were analyzed by immuno-blotting GFP-fusions. * denotes the specific band. **e** Arl5-QL and -TN interact with Lamtor1 but not Lamtor2–5. Bead-immobilized GST-fusion was incubated with cell lysates expressing indicated Myc-tagged Lamtors and pull-downs were analyzed by immuno-blotting Myc-tagged proteins. Lamtor1-Str.-Myc, Lamtor1-G2A-Strep-Myc. **f** Arl5b-TN interacts with Ragulator through Lamtor1. Bead-immobilized GST-Arl5b-TN was incubated with cell lysates expressing indicated Lamtors, and pull-downs were analyzed by immuno-blotting indicated tags. 1 and 2 indicate DMyc-Lamtor5 and DMyc-Lamtor4, respectively. **g** Arl5b-GMPPNP, -GDP, and guanine nucleotide empty form interact with Ragulator. Bead-immobilized GST-Arl5b was first loaded with GMPPNP or GDP or stripped off its bound guanine nucleotide by EDTA treatment. Beads were subsequently incubated with cell lysate, and pull-downs were analyzed by immuno-blotting endogenous Lamtor1–4. **h** Excess Arl5b-TN inhibits the interaction between Rag GTPases and Lamtor1. Cell lysate co-expressing Lamtor1-GFP, Flag-RagB-T54L, and HA-GST-RagC was IPed using anti-GFP antibody in the presence of 0.2 µM recombinant GST-Arl5b-TN or GST. Pull-downs were subsequently immuno-blotted for indicated tags

QL and TN mutant forms of Arl5a, Arl5b or mouse Arl5c (Fig. 5a; Supplementary Fig. 5a, b) localized to the Golgi, although TN form had reduced Golgi localization with concomitantly increased cytosolic pool. In contrast, human Arl5c-wt-GFP did not localize to the Golgi (Supplementary Fig. 5c). We raised Arl5b-specific polyclonal antibody (Supplementary Fig. 5d-f) and the staining of endogenous Arl5b further confirmed its Golgi localization (Fig. 5b). Similar to Arl1[47], the N-terminal myristoylation of Arl5b at Gly of position 2 seemed to be essential for its Golgi localization (Supplementary Fig. 5g). Taking advantage of GLIM (Golgi protein localization by imaging centers of mass), our recently developed quantitative localization method for Golgi proteins[52], localization quotients (LQs) of GFP-tagged Arl5a and b were measured to be $0.99 \pm 0.02$ ($n = 139$) and $0.91 \pm 0.02$ ($n = 94$), respectively, indicating that they mainly localize to the *trans*-Golgi.

Interestingly, in addition to the Golgi pool observed in fixed cells, live-cell imaging revealed that Arl5b-QL-GFP and -TN-GFP also localized to peripheral puncta positive for mCherry-Rab5 (an EE marker), Lamp1-mCherry (a LE or lysosome marker) and Lamtor1-mCherry (Fig. 5c, d), demonstrating the possible endosomal and lysosomal localization of Arl5b. Though peripheral puncta were not observed for Arl5b-wt-GFP under live-cell imaging (Supplementary Fig. 5h), its colocalization with Lamp1 on puncta was revealed by methanol fixation (Supplementary Fig. 5i). The same fixation method also uncovered a small pool of endogenous Arl5b specifically localizing at endolysosomes (Supplementary Fig. 5j). Besides the LE and lysosomal localization of Lamtor1[25,51], as shown by colocalization with GFP-Rab7 and Lamp1, a substantial amount of Lamtor1 also colocalized with EEA1 (an EE marker) (Fig. 5e). However, Lamtor1 did not localize to the Golgi (Supplementary Fig. 5k). Together with our biochemical data, our observation suggests that the interaction between Arl5b and Ragulator can take place on the surface of the endolysosome.

**AA-stimulated retrograde trafficking requires Arl5b and GARP.** Our findings prompted us to test the hypothesis that Arl5b participates in the AA-stimulated endosome-to-Golgi trafficking. Due to the potential redundancy, endogenous Arl5a, b and c were simultaneously depleted by a mixture of three siRNAs targeting the three paralogs (Fig. 5f; Supplementary Fig. 5l, m). The simultaneous depletion of Arl5a and b significantly blunted the AA-stimulated Golgi trafficking of CD8a-furin (Fig. 5g). Single depletion of either Arl5a or Arl5b using alternative shRNAs resulted in similar inhibitory effect (Supplementary Fig. 5n-q). The specificity of Arl5b knockdown was

demonstrated in a rescue experiment by expressing an RNAi-resistant Arl5b (Fig. 5h, j). GARP complex has recently been identified as the effector of Arl5[48]. It localizes to both the TGN and endosomes[53,54] and functions as a tethering factor in the endosome-to-Golgi trafficking[55]. There are four subunits in GARP complex: Vps51–54[55]. Upon depleting endogenous Vps51 or Vps54 (Fig. 5k), the AA-stimulated Golgi trafficking was found substantially attenuated (Fig. 5l). Together, our data demonstrate that Arl5 and its effector, GARP, are essential for the AA-stimulated endosome-to-Golgi trafficking.

**AAs regulate Arl5b–Ragulator interaction.** Interactions between heterodimeric Rag GTPases and components of mTORC1 signaling, including mTORC1, Ragulator, v-ATPase, SLC38A9, and folliculin-FNIP1, are regulated by AAs—they become strengthened and weakened by AA starvation and sufficiency, respectively[20–24,56,57]. We hypothesized that the binding between Arl5b and Ragulator can also be regulated by AAs. Indeed, compared with AA starvation (HBSS treatment), a 20 min stimulation by either AAs or a combination of AAs and serum substantially decreased the amount of Lamtor1 and Lamtor2 co-IPed by Arl5b-GFP (Fig. 6a).

Given our observation that Arl5b-QL interacts with Ragulator more weakly than TN, we initially thought that the reduced Arl5b–Ragulator binding under AA-sufficiency could be due to the guanine nucleotide exchange of Arl5b from GDP to GTP form. Indeed, we later found that AA-sufficiency induces the guanine nucleotide exchange of Arl5b (see below). However, when Arl5b-bound guanine nucleotide was locked by using either QL or TN mutation, reduced-binding under AA-sufficiency was still observed for both mutant forms (Fig. 6a). Therefore, the reduction in Arl5b–Ragulator interaction might be resulted from the structural change in Ragulator upon integrating upstream AA-sufficiency signal from v-ATPase and SLC38A9. Consistent with this view, conA treatment increased Arl5b–Ragulator binding during AA-sufficiency (Fig. 6b). Thus, AAs modulate Ragulator's engagement with Arl5b, similar to previously reported Ragulator–Rag GTPase interaction[24].

Since Gln has the most acute effect on the endosome-to-Golgi trafficking among all AAs, we asked if Gln also plays a significant role in regulating the Arl5b–Ragulator interaction. Following AA starvation (HBSS or DMEM/-AAs treatment), we observed that Arl5b–Lamtor1 interaction was abolished when cells were subsequently treated with Gln alone (Fig. 6c); in contrast, the interaction remained as strong as that in AA starvation when cells were treated with DMEM/-Gln (Fig. 6d). Therefore, Gln is necessary and sufficient to disrupt Arl5b–Ragulator interaction.

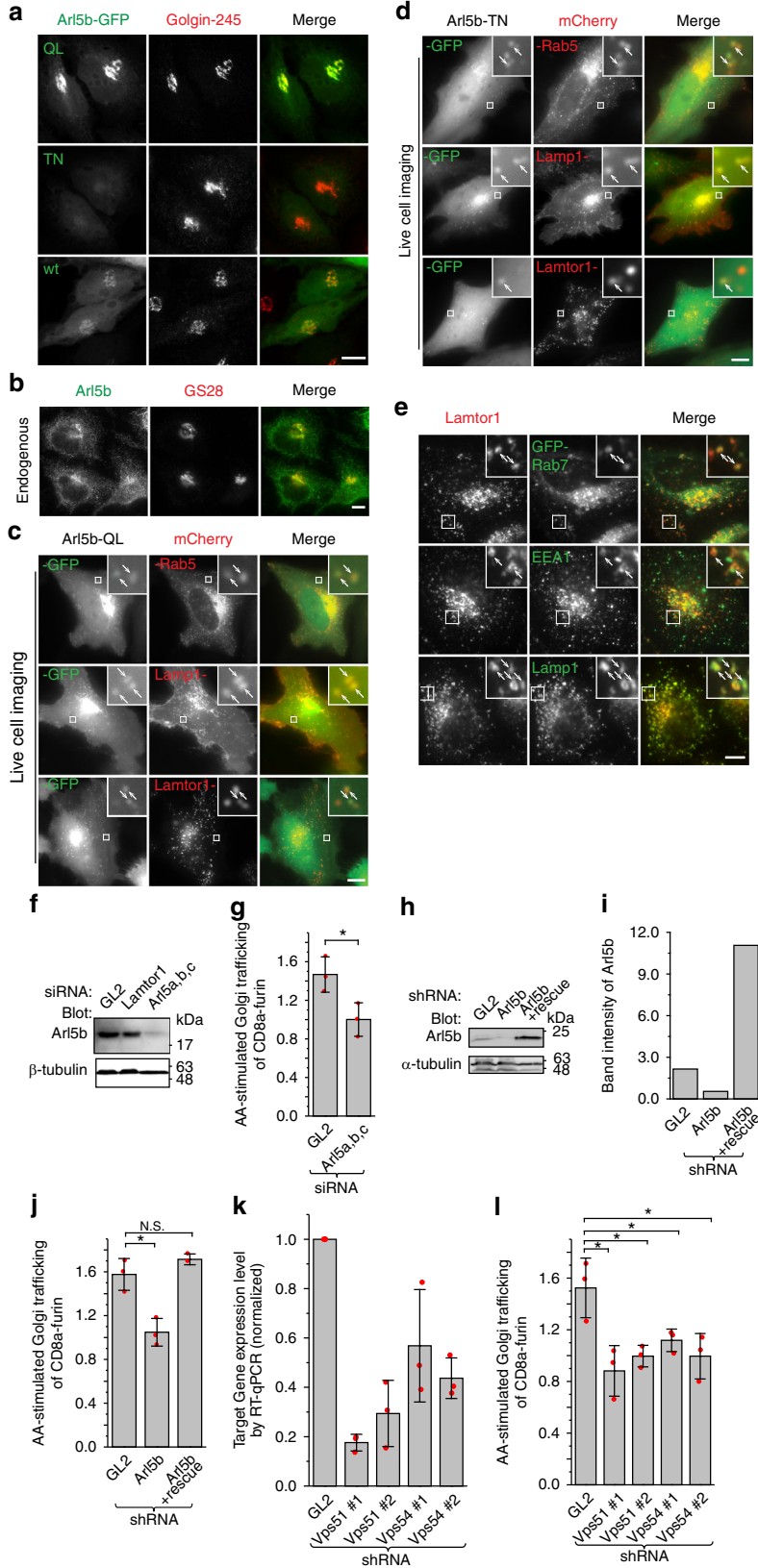

**Ragulator possibly functions as a GEF for Arl5b**. Most small GTPases interact with their effectors in GTP-bound form. Hence it seems unusual that Ragulator preferentially interacts with Arl5b-GDP. Furthermore, we found that Ragulator exhibited higher binding affinity toward guanine nucleotide free Arl5b, which was prepared by ethylenediaminetetraacetic acid (EDTA) treatment, than either Arl5b-GDP or GTP (Fig. 4g). Therefore, the role of Ragulator for Arl5b appears more consistent with a GEF than an effector. In fact, Ragulator was previously demonstrated to be an AA-regulated GEF for small GTPases, RagA and

**Fig. 5** Arl5's localization and its essential role in the AA-stimulated Golgi trafficking. HeLa cells were used. **a** The Golgi localization of different mutant forms of Arl5b. Cells transiently expressing Arl5b-GFP in QL, TN, or wt form were fixed and endogenous Golgin-245 was stained. **b** Endogenous Arl5b localizes to the Golgi. Cells were fixed and endogenous Arl5b and GS28 were co-stained. **c, d** Arl5b colocalizes with Lamtor1 at the endosome and lysosome. Cells transiently co-expressing indicated GFP or mCherry-tagged proteins were imaged under live-cell condition. **e** Lamtor1 localizes to the EE, LE and lysosome. Endogenous Lamtor1 was co-stained with exogenously expressed GFP-Rab7, endogenous EEA1, or Lamp1, respectively. In **c–e**, the boxed region was enlarged in the upper right corner to show the colocalization at puncta (denoted by arrows). Scale bar, 10 μm. **f** The immuno-blot showing that endogenous Arl5b was knocked down by a mixture of siRNAs targeting Arl5a, b, and c. **g** Arl5 is required for the AA-stimulated Golgi trafficking of CD8a-furin. The experiment was conducted as in Fig. 3d. **h–j** When endogenous Arl5b was depleted by lentivirus-transduced shRNA, the expression of an RNAi-resistant Arl5b significantly increased the cellular level of Arl5b and rescued the AA-stimulated Golgi trafficking. **k** The knockdown of endogenous Vps51 and Vps54 by respective lentivirus-transduced shRNAs as assessed by RT-qPCR. The data were from $n = 3$ independent experiments. **l** GARP is required for the AA-stimulated Golgi trafficking. The experiment was conducted as in Fig. 3d. In **g**, **j**, **k**, and **l**, the displayed value is the mean of $n = 3$ independent experiments and individual data points are shown as red dots. Error bar, mean ± s.d.; $P$ values were from $t$ test (unpaired and two-tailed); N.S,. not significant ($P > 0.05$); *$P \leq 0.05$. GL2 is a non-targeting control siRNA or shRNA

RagB[24]. Our findings thus prompted us to test if Ragulator can similarly function as a GEF for Arl5b. The GEF activity was investigated by a fluorescence-based in vitro assay using recombinant components purified from bacteria (Supplementary Fig. 6). We found that, in the presence of Ragulator, the guanine nucleotide exchange of Arl5b was stimulated to >twofold that in the presence of Ragulator subcomplex (Lamtor1–3) or GST (control) (Fig. 6e, f). In cytosol, the guanine nucleotide exchange of Arl5b can couple it to agarose-linked GTP and therefore the amount of Arl5b pulled down by GTP-agarose can be used to monitor the guanine nucleotide exchange in vivo. In control knockdown cells, GTP-agarose pulled down substantially more endogenous Arl5b under AA sufficiency than starvation (Fig. 6g); the AA-stimulated pull-down of Arl5b was blocked by depleting Lamtor1 or SLC38A9 or inhibiting v-ATPase using conA (Fig. 6g, h).We also observed that Gln alone increased while DMEM/-Gln decreased the pull-down (Fig. 6i), suggesting that Gln might be necessary and sufficient for promoting the GEF activity of Ragulator toward Arl5b. Our data therefore suggest that Ragulator might function as a GEF for Arl5b by integrating AA-sufficiency signal from v-ATPase and SLC38A9.

## Discussion
There is a lack of knowledge on how intracellular membrane trafficking processes are regulated in response to extracellular signals. We demonstrate that extracellular AAs, but not growth factors and glucose, can regulate the endocytic membrane trafficking in mammalian cells. Under AA starvation, cargos cycling between the PM and Golgi are arrested in the endosome and the subsequent AA stimulation rapidly promotes the endosome-to-Golgi trafficking of cargos to the Golgi. In response to the availability of AAs, the endosome-to-Golgi trafficking can affect the cellular metabolism in at least two-folds. First, it might prolong half-lives of post-Golgi cycling cargos by diverting them away from lysosomal degradation pathway. Under AA sufficiency, longer half-lives of proteins and lipids probably contribute to anabolic processes for cell growth and proliferation. Second, cells can utilize the AA-stimulated retrograde trafficking to quantitatively adjust cell surface proteins to ensure an optimal response to the environment nutrient. This is reflected by our quantitative proteomics data of surface proteins. In addition to Golgi glycosylation enzymes, endoprotease (furin) and cargo adaptors (sortilin and CI-M6PR), surface localizations of various signaling receptors and secretory ligands are also sensitive to AAs, suggesting nutrient might initiate changes in multiple post-translational modifications, trafficking, and signaling pathways.

We observed that Gln, one of the most effective nitrogen sources for yeast, most acutely stimulates the endocytic pathway. Gln has the highest concentration in both blood plasma (0.5–0.8 mM) and cell culture media (2–4 mM). It fuels the tricarboxylic acid cycle and contributes to the biosynthesis of macromolecules; furthermore, it facilitates the uptake of essential AAs and activates mTORC1 signaling pathway[58,59]. In fact many cancer cells heavily rely on Gln for their growth and survival, the complete mechanism of which remains to be elucidated. It is possible that the maintenance of the endocytic retrograde trafficking and the presentation of certain cell surface proteins contributes to the cellular demand for Gln.

We elucidated a signaling pathway from the sensing of AAs to the trafficking of membrane carriers from the endosome to the Golgi. The AA-regulated retrograde trafficking and mTORC1 signaling share common components, including SLC38A9, v-ATPase, and Ragulator, but the two pathways diverge after Ragulator: while the activation of mTORC1 is through Rag GTPases, the promotion of the retrograde trafficking requires the Arl-subfamily small GTPase—Arl5b. The Golgi-localized Arl5b has been previously known to participate in the endosome-to-Golgi trafficking by interacting with the tethering complex, GARP[48,49]. We further discovered that Ragulator interacts with Arl5b on the endolysosome and Arl5b and GARP are essential players for the AA-stimulated endosome-to-Golgi trafficking. We found that AAs stimulate the guanine nucleotide exchange of Arl5b from GDP to GTP in a Ragulator, v-ATPase and SLC38A9 dependent manner. Our in vitro assay provides an evidence that Ragulator could function as a GEF for the activation of Arl5b, therefore making a mechanistic connection between endocytic trafficking and the nutrient signaling pathway. We propose here a working model to summarize the AA-regulated signaling pathway that leads to the retrograde trafficking (Fig. 6j). Similar to mTORC1 signaling pathway, luminal AAs of the endolysosome are first sensed by SLC38A9 and v-ATPase, which in turn signal to Ragulator; activated Ragulator subsequently acts as a GEF to activate Arl5b; at last, GARP is recruited to the membrane carrier by activated Arl5b and facilitates the tethering and fusion of the budded membrane carriers with the TGN membrane. It is unknown at this moment if endolysosome-localized Ragulator can contribute to the activation of Arl5b on the Golgi membrane. In agreement with its possible function as a GEF, four subunits of Ragulator, Lamtor2–4, have roadblock domains, which are expected to have GTPase-binding activity[60].

The endosome-to-Golgi trafficking has been established to regulate physiological and pathological processes such as metazoan development, toxin invasion, cellular homeostasis and neurological diseases[1,6]. Our discovery therefore implies that nutrient might modulate and affect these processes, an interesting prediction awaiting further study.

## Methods
**Antibodies and small molecules**. The following antibodies were purchased from Cell Signaling Technology: rabbit anti-p70S6K polyclonal antibody (pAb) (#9202)

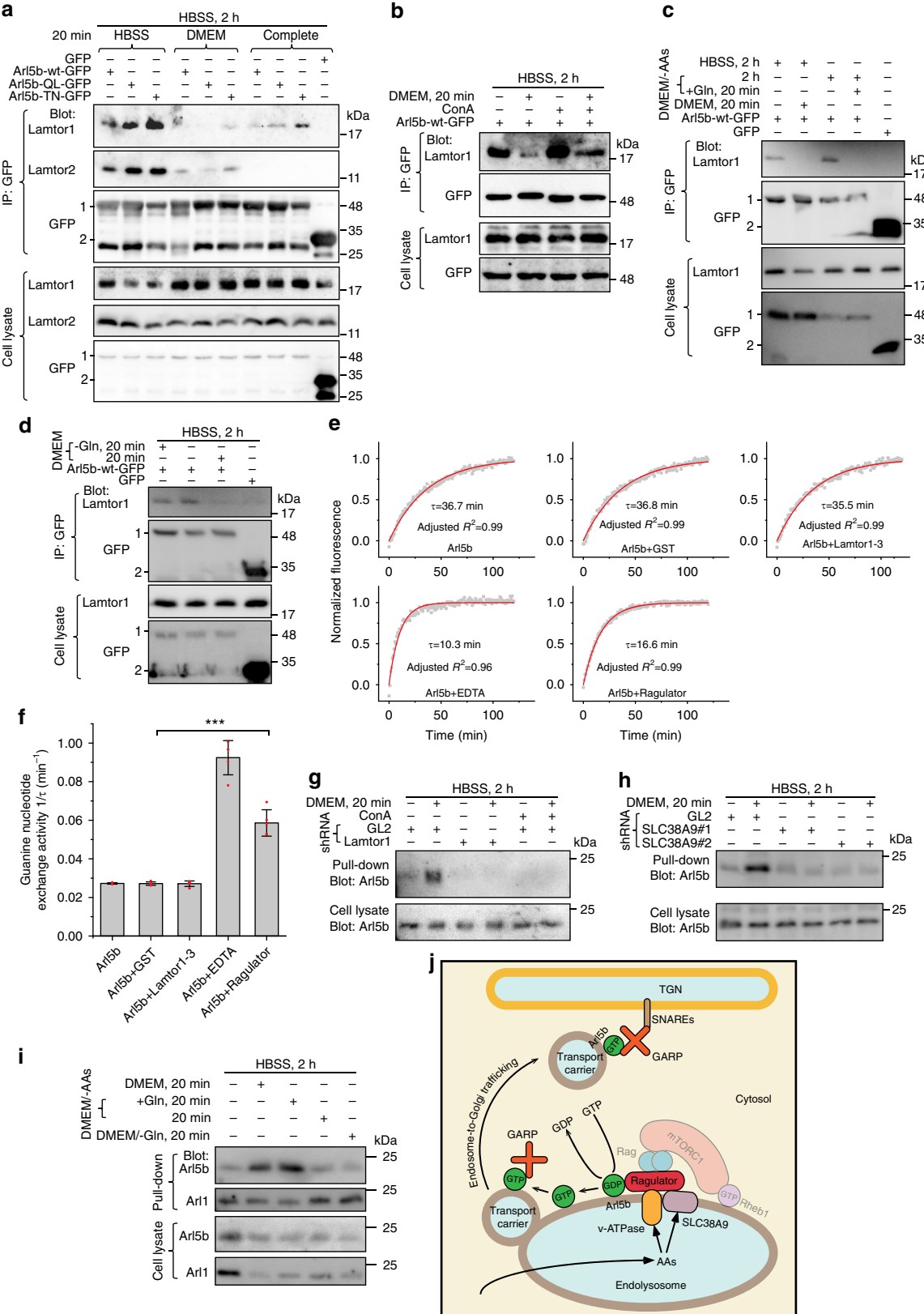

(1:1000 for Western blot or WB), rabbit anti-RagA monoclonal antibody (mAb) (#4357)(1:1000 for WB), rabbit anti-Lamtor1 mAb (#8975)(1:1000 for WB; 1:200 for immunofluorescence or IF), rabbit anti-Lamtor2 mAb (8145) (1:1000 for WB), rabbit anti-Lamtor3 mAb (8168) (1:1000 for WB), and rabbit anti-Lamtor4 mAb (#13140) (1:1000 for WB). The following antibodies were from Abcam: rabbit anti-

α-tubulin pAb (#4074) (1:1000 for WB), rabbit anti-EEA1 pAb (#ab2900) (1:1000 for WB; 1:500 for IF), rabbit anti-Giantin pAb (#ab24586)(1:1000 for IF), rabbit anti-GM130 pAb (#ab52649)(1:500 for IF), rabbit anti-TGN46 pAb (#50595)(1:400 for IF), and HRP-conjugated protein A (#ab7456)(1:3000 for WB). The following antibodies were from Santa Cruz: rabbit anti-GAPDH pAb (#sc-

**Fig. 6** Regulator likely functions as a GEF for Arl5b downstream of v-ATPase and SLC38A9. **a** AAs disrupt Arl5b–Ragulator interaction, regardless of guanine nucleotide binding status of Arl5b. HEK293T cells expressing indicated GFP-fusions were starved in HBSS for 2 h before treatment by indicated medium for 20 min. The resulting cell lysates were incubated with anti-GFP antibody and co-IPs were analyzed by immuno-blotting indicated proteins. **b** AA-induced disruption of Arl5b–Ragulator binding is inhibited by conA. HEK293T cells expressing Arl5b-wt-GFP were treated by HBSS for 2 h. Cells were lysed directly or after 20 min treatment of DMEM. For conA treatment, 2.5 µM conA was used throughout the incubation. Co-IPs and immuno-blotting were performed as in **a**. **c** Gln is sufficient to disrupt Arl5b-Ragulator binding. HEK293T cells expressing Arl5b-wt-GFP or GFP were starved in HBSS or DMEM/-AAs for 2 h and subsequently incubated with indicated medium. Co-IPs and immuno-blotting were performed as in **a**. **d** Gln is necessary to disrupt Arl5b–Ragulator binding. The experiment was performed as in **c**. **e**, **f** Ragulator, but not its subcomplex containing Lamtor1–3 can stimulate the guanine nucleotide exchange of Arl5b in vitro. **e** Typical exchange kinetic traces of Arl5b. Guanine nucleotide exchange activities under different factors are quantitatively expressed by $1/\tau$ in **f**. Red dots, individual data points; error bar, mean ± s.d.; P values were from t test (unpaired and two-tailed); ***$P \leq$ 0.0005. **g**, **h** v-ATPase, Ragulator and SLC38A9 are essential for the AA-stimulated guanine nucleotide exchange of Arl5b. HeLa cells were subjected to shRNA-mediated knockdown of indicated proteins. Nutrient treatment was performed as in **d**. ConA treatment was performed as in **b**. The resulting cell lysates were incubated with GTP-agarose and pull-downs were analyzed by immuno-blotting endogenous Arl5b. GL2 is a non-targeting control shRNA. **i** Gln is necessary and sufficient for the AA-stimulated guanine nucleotide exchange of Arl5b. Nutrient treatment and pull-downs were performed as in **g** and **h**. In **a**, **c**, and **d**, 1 and 2 indicate Arl5b-(wt, QL or TN)-GFP or GFP band, respectively. **j** A working model on how Arl5b integrates the AA-sufficiency signal and regulates the endosome-to-Golgi trafficking. See discussion for details

25778)(1:1000 for WB), mouse anti-GFP mAb (#sc-9996)(1:1000 for WB), mouse anti-Myc mAb (#sc-40) (1:1000 for WB), mouse anti-HA mAb (#sc-7396)(1:1000 for WB), and rabbit anti-Rab7 pAb (#sc-10767) (1:1000 for WB). Mouse anti-Lamp1 mAb (H4A3)(1:500 for IF) and mouse anti-CD8a mAb (OKT8)(1:500 for IF) were from Developmental Studies Hybridoma Bank. Rabbit anti-furin pAb (#PA1062)(1:1000 for WB; 1:100 for IF), mouse anti-CI-M6PR mAb (#MA1066) (1:200 for IF), Alexa Fluor 594 conjugated Cholera toxin B fragment (#34777) (1:300 for the endocytic trafficking assay), Alexa Fluor conjugated goat anti-mouse (1:500 for IF), and anti-rabbit IgG antibodies (1:500 for IF) were from Thermo Fisher Scientific. Mouse anti-Flag mAb was from Sigma-Aldrich (#F1804)(1:1000 for WB). HRP-conjugated goat anti-mouse (#176516)(1:10,000 for WB) and anti-rabbit IgG antibodies (#176515)(1:10,000 for WB) were from Bio-Rad. Mouse anti-GM130 mAb (#610823)(1:500 for IF), mouse anti-EEA1 mAb (#610456)(1:500 for IF), and mouse anti-syntaxin6 mAb (#51-9002100)(1:1000 for WB) were from BD Biosciences. Rabbit anti-RUFY1 pAb (#13498-1-AP)(1:250 for IF) was from Proteintech. Rabbit anti-Arl5b antibody (1:500 for WB; 1:100 for IF) is generated as described below. Rabbit anti-Arl1 pAb (1:1000 for WB) was previously prepared[61].

The following small molecule inhibitors are commercially available: conA (Abcam, #ab144227); Torin1 (Tocris Bioscience, #4247); and rapamycin (InvivoGen, #tlrl-Rap). GMPPNP (#G0635) and GDP (#G7127) were from Sigma-Aldrich.

**Yeast two-hybrid screening.** AH109 yeast cells harboring Arl5b-QL in pGBKT7 vector were mated to Y187 yeast cells pre-transformed with human kidney cDNA library (Clontech). The resulting diploid yeast cells were selected on synthetic drop out medium without Trp, Leu, His and Ade. Gal4-activation-domain-fused cDNAs were subsequently extracted from positive yeast clones and identified by DNA sequencing.

**Cell culture and transfection.** HeLa, BSC-1, and HEK293T cells were from American Type Culture Collection. 293FT cells were from Thermo Fisher Scientific. Cells were maintained in high glucose DMEM (GE Healthcare Life Sciences) supplemented with 10% fetal bovine serum (FBS) (Thermo Fisher Scientific) at 37 °C in 5% $CO_2$ incubator. Live-cell imaging of HeLa cells was performed in $CO_2$ Independent Medium (Thermo Fisher Scientific) supplemented with 4 mM Gln and 10% FBS at 37 °C. HeLa, BSC-1, and HEK293T cells were transfected using polyethylenimine (Polysciences Inc.). Transfection was performed when cells reached 70–80% confluency according to standard protocol.

DMEM-base was prepared using 100× MEM vitamin solution (Thermo Fisher Scientific, #11120052), inorganic salts, glucose, and sodium pyruvate according to the formulation of DMEM from Thermo Fisher Scientific (#11965) leaving out all AAs. Selective AA(s) was(were) added to DMEM-base to make corresponding media containing defined AAs. DMEM/-Gln and DMEM/-Leu were prepared by supplying Leu and Gln, respectively, to DMEM/-Gln/-Leu (MP Biomedicals, #1642149). HBSS was prepared according to the formulation of Thermo Fisher Scientific HBSS (#14025126). Except Gln (Thermo Fisher Scientific) and His (Fluka), all AAs were from Sigma-Aldrich. Concentrations of individual AAs in nutrient media were either according to the formulation of DMEM of Thermo Fisher Scientific (#11965) or as indicated in the text. Dialyzed serum was prepared by dialyzing the serum in 3.5 kDa molecular weight cut-off dialysis tubing (Thermo Fisher Scientific, #68035) against phosphate-buffered saline (PBS) followed by passing through a syringe-driven 0.22 µm filter unit (Sartorius).

**Surface labeling.** Surface labeling was conducted by incubating live cells with anti-CD8a antibody (OKT8) for 1 h on ice. Un-bound antibody was subsequently washed away by ice cold PBS and cells were incubated in AA-starvation or

-sufficiency medium at 37 °C for certain length of time before being processed for imaging. Acid wash was conducted to strip-off surface-exposed CD8a antibody that binds to CD8a-furin. Briefly, live cells were incubated with ice cold 0.2 M acetic acid in 0.5 M NaCl for 4 min and subsequently washed extensively by ice cold PBS. Cells were then subjected to endocytic trafficking at 37 °C in indicated medium.

To label surface and intracellular pools of CD8a-chimeras, transfected HeLa cells were first treated with DMEM or HBSS for 2 h. In Fig. 2j experiment, cells were subsequently subjected to surface labeling by anti-CD8a antibody followed by fluorescence-conjugated secondary antibody. Next, after fixation and permeabilization, cells were stained by anti-CD8a antibody followed by another fluorescence-conjugated secondary antibody to label intracellular pool of CD8a-chimera. In Fig. 3i experiment, only surface CD8a-furin-mEos2 was fluorescence-labeled while the intracellular pool was quantified by the total intensity of mEos2.

**Golgi-to-PM transport assay.** The Golgi-to-PM transport assay of furin was performed using the retention using selective hooks reporter system. HeLa cells transiently expressing the reporter construct, ss-Strep-KDEL_ss-SBP-GFP-CD8a-furin[52], were treated with DMEM or HBSS together with 50 µM biotin and 10 µg ml$^{-1}$ cycloheximide at 20 °C for 2 h to accumulate the reporter at the Golgi. Cells were subsequently warmed up to 37 °C to chase the Golgi-to-PM trafficking of SBP-GFP-CD8a-furin. The arrival at the PM was quantified by surface staining of GFP and normalized by the total cellular GFP intensity.

**Lentivirus-mediated knockdown and expression.** 293FT cells seeded in a six-well-plate were transfected with shRNA in pLKO.1 vector or a mixture of pLVX expression construct, pLP1, pLP2, and pLP/VSVG using Lipofectamine 2000 (Thermo Fisher Scientific). Cells were incubated at 37 °C for 18 h and replaced with fresh medium to incubate for another 24–48 h to harvest virus-containing supernatant. The supernatant was passed through 0.45 µm filter (Sartorius) and used immediately. For lentivirus-mediated transduction, HeLa cells were incubated with the virus supernatant in the presence of 8 µg ml$^{-1}$ of polybrene (Sigma-Aldrich, #H9268) 24–48 h before an assay. For the expression of an exogenous protein after knockdown, transient transfection was conducted 48 h after the addition of shRNA expressing lentivirus and cells were further incubated for 24 h before the indicated assay. For the rescue experiment, after Lamtor1 or Arl5b shRNA mediated knockdown, HeLa cells were incubated with RNAi-resistant Lamtor1 or Arl5b expressing lentivirus in the presence of polybrene for 24–48 h before further transfection with CD8a-furin. To select CD8a-furin stable cells, HeLa cells infected by lentivirus carrying CD8a-furin in pLVX-puro vector were cultured in the medium containing 1 µg ml$^{-1}$ puromycin (Sigma-Aldrich). The pooled stable cells were subjected to limited dilution and cultured in a 96-well plate to expand single cell colonies. The colonies were further screened by immunofluorescence or western blot.

**RNAi-mediated knockdowns.** The following siRNAs were purchased from Dharmacon Inc: luciferase GL2 (#D-001100-01-20) and siRNA SMART pools for human Lamtor1 (#L-020916-02-0005), human Arl5a (#L-012408-00-0005), Arl5b (#L-017861-02-0005) and Arl5c (#L-030887-02-0005). siRNAs were transfected to HeLa cells using Lipofectamine 2000 according to manufacturer's protocol. For the expression of an exogenous protein after knockdown, transfections were conducted 24 h after siRNA transfection. 48 h after the transfection of siRNA, cells were processed for assays.

**RT-qPCR.** Total RNA was purified using TRIzol$^{TM}$ reagent (Thermo Fisher Scientific) from HeLa cells according to standard protocol. Reverse transcription primed by random nonamer primers was conducted using nanoScript 2 Reverse

Transcription kits (Primerdesign). The real-time PCR was subsequently performed on a Bio-Rad CFX96 Touch™ real-time PCR detection system using SYBR green based PrecisionFAST with LOW ROX qPCR kit (Primerdesign). The specificity of PCR primers were verified by both melt curves and agarose gel electrophoresis. The primer pairs are listed below: Arl5a (5′-GTT AGC GCA TGA GGA CCT AAG-3′, 5′-CTT GGC ACA ATC CCT CGC CAG TTA C-3′), Arl5b (5′-TGG CTC ATG AGG ATT TAC GGA AG-3′, 5′-CCT TGG CAT AAC CCT TCT CCT GTG-3′), hArl5c (5′-TGG CCC ATG AGG CTC TAC AGG ATG-3′, 5′-TCC ATC CAC TGA AGT CTG GCA G-3′), Vps51 (5′-CTC AGC CAC AGA CAC CAT CCG G-3′, 5′-GCG AGC GCT GAA GTC GGT GAT C-3′), Vps54 (5′-GTT GTT GTG AAG CTT GCA GAT CAG-3′, 5′-TGT TGC CTT CAC TCT CTG TAG G-3′), SLC38A9 (5′-CCT AGC ATT TTC CAT GTG CTG-3′, 5′-GCT CCT GAA TAT CTT ATG ATC CCT CC-3′) and Lamtor3 (5′-CCT GTT ATT AAA GTG GCA AAT GAC AAT GC-3′, 5′-TTG AAC CAC CTG GTA GGT GTT ATA G-3′).

**Subcellular fractionation by sucrose gradient.** HeLa cells stably expressing CD8a-furin were cultured in two Φ15 cm Petri-dishes to 70% confluency and treated with HBSS for 2 h. Cells on one plate were further incubated with DMEM for 20 min. Next, cells on both plates were lysed in 10% sucrose buffer (10% sucrose, 3 mM imidazole, pH 7.4, 1 mM EDTA) by repeatedly extruding them through a 25 1/2 Gauge needle. The cell lysates were subsequently centrifuged at $1000 \times g$ for 10 min. The resulting two supernatants were separately loaded on the top of two tubes containing 10–40% continuous sucrose gradient, which were then subjected to ultracentrifuge in SW28 rotor (Beckman) at $140,000 \times g$ and 4 °C for 5 h. After the centrifugation, samples in tubes were collected into 20 fractions with 1.85 ml per fraction. Proteins within each fraction were pelleted down using methanol/chloroform method[62], dissolved in SDS-sample buffer and analyzed by western blot.

**Preparation of Cy5-conjugated STxB.** Escherichia coli cells harboring plasmid pSTxB(sulf)$_2$[63] were cultured at 30 °C and subjected to heat shock at 42 °C to induce the expression of STxB at the periplasm. At room temperature, cells were subjected to buffer 1 (10 mM Tris pH 8.0) for 10 min and buffer 2 (25% sucrose, 1 mM EDTA, 10 mM Tris, pH 8.0) for 10 min. Next, cells were pelleted and re-suspended in ice cold water for 10 min. After centrifugation, the supernatant was passed though Q-Sepharose column (GE Healthcare Life Sciences) to obtain purified STxB. The purified STxB was conjugated to Cy5 using cyanine5 NHS ester (Lumiprobe, #13020).

**Purification of GST-tagged fusion proteins.** Plasmid constructs encoding GST-tagged fusion proteins were transformed into BL21 E. coli cells. After induction by 0.25 mM Isopropyl β-D-1-thiogalactopyranoside (IPTG), bacterial pellets were lysed by sonication in lysis buffer containing 50 mM Tris pH 8.0, 100 mM NaCl, 1% Triton X-100, 1 mM dithiothreitol (DTT), and 1× cOmplete™ Protease Inhibitor Cocktail (Roche). The supernatant collected after high-speed centrifugation was incubated with Glutathione Sepharose 4B beads (GE Healthcare Life Sciences) for 4–12 h at 4 °C cold room. After extensive washing using a buffer containing 50 mM Tris pH 8.0, 100 mM NaCl, and 0.1% Triton X-100, beads with immobilized GST-fusion proteins were used for pull-downs or cross-linking followed by antibody purification. Alternatively, the immobilized fusion protein was eluted using 10 mM reduced glutathione dissolved in a buffer containing 50 mM Tris pH 8.0 and 100 mM NaCl.

**Purification of His-tagged fusion proteins.** Plasmid constructs encoding His-tagged fusion proteins were transformed into BL21 E. coli cells. After induction by 0.25 mM IPTG, bacterial pellets were lysed by sonication in lysis buffer containing 100 mM HEPES pH 8.0, 500 mM KCl, 1% Triton X-100, 1 mM β-mercaptoethanol, and 1× cOmplete™ EDTA-free Protease Inhibitor Cocktail (Roche). The supernatant collected after high-speed centrifugation was added imidazole to a final concentration of 5 mM and subsequently incubated with nickel-nitrilotriacetic acid agarose (Qiagen), which was pre-washed in a buffer containing 100 mM HEPES pH 8.0, 500 mM KCl and 10 mM imidazole for 4–12 h at 4 °C cold room. After extensive washing using a buffer containing 20 mM HEPES pH 8.0, 200 mM KCl, 10% glycerol and 25 mM imidazole, bead-immobilized His-tagged fusion proteins were eluted in a detergent-free elution buffer containing 20 mM HEPES pH 8.0, 200 mM KCl, 10% glycerol, and 250 mM imidazole.

**Purification of Ragulator and Ragulator subcomplex.** His-tagged Lamtor1[64], cloned in pETDuet-1 (Novagen), non-tagged Lamtor2–3[64], cloned in dual-expression vector pACYCDuet (Novagen), and non-tagged Lamtor4–5[64], cloned in dual-expression vector pRSFDuet (Novagen) were co-expressed in BL21 E. coli cells and purified as a His-tagged fusion protein to yield Ragulator. To purify Ragulator subcomplex containing Lamtor1–3, above mentioned plasmids encoding Lamtor1 and Lamtor2–3 were used to co-transform BL21 E. coli cells.

**In vitro guanine nucleotide exchange assay.** All purified proteins were quantified for their concentrations and subjected to Zeba Spin Desalting Column (molecular weight cut-off 7 kDa) (Thermo Fisher Scientific) to change their buffers

to HK buffer containing 50 mM HEPES pH 7.5, 120 mM KCl and 1 mM DTT. Guanine nucleotide exchange reactions were assembled in a black 96-well plate (Greiner Bio-One) in 150 μl HK buffer containing additional 1 mM MgCl$_2$, 1 μM Mant-GMPPNP (Thermo Fisher Scientific), 1 μM His-Δ14Arl5b and 0.7 μM EDTA, Ragulator or Ragulator subcomplex. The change of fluorescence was monitored in Cytation 5 (BioTek) with excitation at 360 nm and emission at 440 nm. The fluorescence data were collected every 50 s and subsequently subjected to single exponential curve-fitting ($y = y_0 + A * \exp(-(x-x_0)/\tau)$) in OriginPro2015 (Origin Lab). All fitted data had adjusted-$R^2 \geq 0.95$.

**Generation of anti-Arl5b rabbit polyclonal antibody.** The plasmid construct encoding Arl5b-His (Arl5b in pET30a) was transformed into BL21 E. coli cells. After induction by 0.25 mM IPTG, bacterial pellet was lysed by sonication in PBS containing 8 M urea. The supernatant collected after high-speed centrifugation was incubated with nickel-nitrilotriacetic acid agarose at room temperature for 2 h. Beads were washed with PBS containing 8 M urea and 20 mM imidazole and the bound Arl5b-His was eluted in PBS containing 8 M urea and 250 mM imidazole. After concentrating and changing the buffer to PBS containing 4 M urea, Arl5b-His was used to immunize rabbits, and anti-sera were collected by Genemed Synthesis Inc.

To purify polyclonal antibody against Arl5b, GST-Arl5b immobilized on Glutathione Sepharose 4B beads was incubated with 50 mM Dimethyl pimelimidate in 200 mM sodium borate pH9.0 to cross-link GST-Arl5b onto glutathione beads. After washing with 200 mM ethanolamine pH 8.0, GST-Arl5b-cross-linked beads were incubated with anti-serum at room temperature for 1 h and washed by PBS. The antibody bound to beads was eluted by 100 mM glycine pH 2.8 and immediately neutralized using minimal amount of 1 M Tris pH 8.0 followed by dialysis against PBS.

**Guanine nucleotide exchange of GST-Arl5b.** Glutathione bead-immobilized GST-Arl5b was washed twice with the exchange buffer (20 mM HEPES pH 7.4, 100 mM NaCl, 10 mM EDTA, 5 mM MgCl$_2$ and 1 mM DTT) and incubated with the exchange buffer supplemented with 10 unit ml$^{-1}$ calf intestinal alkaline phosphatase (New England Biolab) at room temperature for 2 h. Next, beads were washed by the exchange buffer and incubated with the same buffer supplemented with 0.5 mM GMPPNP or GDP (final concentration) for 1 h at the room temperature. 5 mM (final concentration) MgCl$_2$ was subsequently added to the system and beads were further incubated for 1 h at the room temperature. The exchanged GST-Arl5b on beads was stored at 4 °C until use.

**Co-IP and pull-down.** HEK293T or HeLa cells were transfected by indicated DNA construct to express the exogenous protein and/or treated with the AA-starvation or -sufficiency medium as indicated in the text. After washing cells with ice cold PBS, cells were lysed in lysis buffer (40 mM HEPES, pH 7.4, 150 mM NaCl, 1% Triton X-100, 2.5 mM MgCl$_2$, and 1 mM PMSF) and cleared by centrifugation at 16,000×g for 30 min. In co-IPs or pulldowns involving Ragulator, Triton X-100 was substituted by 0.3% CHAPS (3-[(3-Cholamidopropyl)dimethylammonio]-1-pro-panesulfonate, Sigma-Aldrich). Cell lysates were subsequently incubated with ~1 μg antibody against the protein of interest, 15 μl GFP-Trap beads (ChromoTek), or 10–40 μg GST-fusion protein immobilized on glutathione beads for 4–14 h in a cold room. When antibody was used, the antigen–antibody complex was subsequently captured by 15 μl pre-washed Protein A/G beads (Pierce). After washing beads extensively with the lysis buffer, bound proteins were eluted by boiling in SDS-sample buffer and resolved in 8–12% SDS-PAGE. Western blot was subsequently conducted to detect bound proteins according to standard protocol. Separated proteins were transferred to polyvinyl difluoride membrane (Bio-Rad). After primary and HRP-conjugated secondary antibody incubation, the chemiluminescence signal was detected by a cooled charge-coupled device camera (LAS-4000, GE Healthcare Life Sciences). Uncropped blot images are presented in Supplementary Fig. 7.

**GTP-agarose pull-down.** HeLa cells suspended in binding buffer (20 mM HEPES, pH 8.0, 150 mM NaCl, 10 mM MgCl$_2$ and 1× cOmplete™ EDTA-free Protease Inhibitor Cocktail) were lysed by repeatedly extruding them through a 25 1/2 Gauge needle. Lysates were cleared by centrifugation at 16,000×g for 30 min and then subjected to incubation with GTP-agarose beads (bioWORLD) for 1 h at 4 °C. Beads were washed for three times with binding buffer and bound proteins were eluted by boiling in SDS-sample buffer and analyzed by western blot.

**Immunofluorescence labeling.** Cell seeded on Φ 12 mm glass coverslips were fixed by 4% paraformaldehyde in PBS at room temperature for 20 min. After neutralizing paraformaldehyde with 100 mM ammonium chloride, cells were washed by PBS and incubated with primary antibody diluted in antibody dilution buffer, which is PBS supplemented with 5% FBS, 2% bovine serum albumin, and 0.1% Saponin (Sigma-Aldrich). Cells were subsequently washed by PBS and incubated with fluorescence-conjugated secondary antibody diluted in antibody dilution buffer. After PBS washing, the coverslip was mounted in Mowiol 4-88 (EMD Millipore). By default, cells were fixed by paraformaldehyde. For methanol fixation, cells were immersed in −20 °C methanol for 5 min followed by another 5

min of methanol treatment at room temperature. After extensive PBS washing, cells were processed for immunofluorescence labeling as normal.

**Fluorescence microscopy**. High resolution images were acquired under an inverted wide-field microscope system, comprising Olympus IX83 equipped with a Plan Apo oil objective lens (63× or 100×, NA 1.40), a Plan Apo dry objective lens (20×, NA 0.75), a motorized stage, motorized filter cubes, a scientific complementary metal-oxide semiconductor camera (Neo; Andor) and a 200 W metal-halide excitation light source (Lumen Pro 200; Prior Scientific). Dichroic mirrors and filters in filter turrets were optimized for GFP/Alexa Fluor 488, mCherry/Alexa Fluor 594 and Alexa Fluor 647. The microscope system was controlled by Meta-Morph software (Molecular Devices) and only center quadrant of the camera sensor was used for imaging.

**Image quantification**. For imaging analysis, random fields of view were imaged. In each image, border-cells were excluded and all the rest cells expressing reporters were analyzed. Image analysis was performed in ImageJ (http://imagej.nih.gov/ij/). In transient transfection, cells have various levels of expression of the reporter. Therefore, different cells should have distinct cellular background fluorescence intensities. The region of interest (ROI) of the cell was manually drawn by tracing the cell's contour. The ROI of the Golgi was generated by intensity thresholding using the co-stained endogenous Giantin or Golgin-245 signal. The image is background-subtracted using ROIs outside cells. In the channel of the reporter fluorescence, $A_{cell}$ and $A_{Golgi}$ are the area (in pixels) of the cell and the Golgi ROI respectively, while $I_{cell}$ and $I_{Golgi}$ are the mean intensity of the cell and the Golgi ROI respectively. $f$ is a constant value between 0 and 1. $f = 0.5$ was used for image quantification with either transfected or endogenous reporters. The fraction of Golgi-localized reporter was calculated as $(I_{Golgi} - f^*I_{cell})^*A_{Golgi}/((1-f)^*I_{cell}^*A_{cell})$.

To calculate AA-stimulated Golgi trafficking, the mean fraction of Golgi-localized CD8a-furin under AA-stimulation was divided by that under AA starvation. In panels a, d, f, h and j of Fig. 3 and panels g, j and l of Fig. 5, three independent experiments were performed and minimal numbers of cells analyzed are 78, 21, 61, 15, 50, 49, 30, and 29, respectively. For Fig. 3i experiment, the normalized surface intensity of each cell was first calculated as the total surface intensity of CD8a-furin-mEOS2 divided by mEOS2 total cellular intensity. To acquire the surface DMEM/HBSS-ratio, the mean normalized surface intensity under DMEM was divided by that under HBSS. Three independent experiments were performed and the minimal number of cells analyzed was 86.

**Statistics**. *P* values were determined using Student's *t* test (unpaired and two-tailed).

## Data availability

The mass spectrometry proteomics data have been deposited to the Proteo-meXchange Consortium via the PRIDE partner repository with the dataset identifier PXD011136. Data that support the findings of this study are available upon request to the corresponding author.

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

## Acknowledgements

We would like to thank W. Hong lab (Institute of Molecular and Cell Biology, Singapore) for the initial support of this work, B. Wu (Nanyang Technological University, Singapore) for initial trials in the purification of Ragulator and D. Sabatini, T. Kirchhausen, D. Root, and S. Manley for sharing DNA plasmids. This work was supported by the following grants to L.L.: NMRC/CBRG/007/2012, MOE AcRF Tier1 RG132/15, Tier1 RG35/17, Tier1 RG48/13, and Tier2 MOE2015-T2-2-073.

## Author contribution

L.L. conceived, designed, and supervised the study. M.S., B.C., D.M., B.K.B., Y.Z., and H.C.T. designed and performed experiments. B.D. and S.K.S. helped in mass spectrometry and analysis. G.W. helped in the preparation of Ragulator. M.S., B.C., and L.L. analyzed the rest data. L.L. wrote the manuscript.

## Additional information

**Competing interests:** The authors declare no competing interests.

