## [Peer Review File · Nature Communications]

Reviewers' comments:

Reviewer #1 (Remarks to the Author):

This is an intriguing study proposing that amino acids (AA) regulate retrograde transport between the TGN and endosomes by a pathway involving SLC38A9, v-ATPase, Ragulator, Arl5 and GARP, but not the Rags or mTORC1. It is certainly of great interest that AA levels would regulate retrograde transport in addition to their well-established roles in regulating anabolic and catabolic pathways. However, some of the evidence in this paper is not compelling enough to support such an important conclusion.

Specific comments

1. The claim that Ragulator functions as a GEF for Arl5 is based on a poorly validated assay involving binding to GTP-agarose. More conventional GEF assays should be used, such as in vitro exchange of GTP for GDP using purified Ragulator and Arl5.
2. Line 57. Has mammalian GARP been shown to mediate sorting at the endosomal membrane?
3. Line 140 and Fig. 1h,i. The gradient does not support the reduced localization of CD8-furin to the TGN, certainly not as much as shown in the IFs in panels e-g. Similarly, the sentence in line 150 "The reduction of the Golgi pool..." is not supported by the gradient data.
4. The redistribution of endogenous CI-MPR shown in Suppl. Fig. 1c is not convincing, since giantin also seems to change in starved cells.
5. Line 188 and Fig. 2f. Glu is as effective as Gln, but this fact is not discussed. Does it make sense in light of other metabolic regulation pathways?
6. Line 259. If Lamtor1 or Ragulator is a GEF for Arl5, it doesn't make sense that it interacts with GTP- or -GDP-loaded, but not WT, Arl5 (Fig. 4a). The same applies to mutant Arl5 constructs. Why would it interact at all with the GTP-locked form, and even better than with WT Arl5? If anything, a GEF should interact with GDP>WT>GTP forms.
7. Line 270 and Fig. 4d. Lamtor1 is a well folded protein, N- and C-terminal truncations can cause misfolding. I'm not sure the results are valid.
8. Ragulator is an abundant complex, so it should be possible to perform GST-Arl5 pulldowns with the endogenous complex. Are all Ragulator subunits pulled down under these conditions?
9. Line 335 and Fig. 3i. Shouldn't the knock down to GARP subunits inhibit retrograde transport irrespective of AA levels? Why only AA-stimulated retrograde transport?
10. Do AA levels regulate Arl5 and GARP recruitment to membranes?
11. Line 343. The statement "it becomes strengthened..." is wrong. It's just the opposite.
12. Fig. 3. I am concerned that the "AA-stimulated Golgi trafficking of CD8-furin" values for the control samples range from 1.2 to 2.0 in the different experiments. How significant can then differences between 1.6 (control) and 1.2 (Lamtor3 KD) be in a set of experiments? The same applies to other experiments.

Reviewer #2 (Remarks to the Author):

The paper by Shi et al identifies a connection between an amino acid-sensing complex located at the lysosome and the regulation of a retrograde trafficking pathway that sorts membrane proteins from endosomes back to the plasma membrane via the Golgi. The authors show that the Lamtor complex, which is involved in mTORC1 regulation by amino acids, may be a guanine nucleotide exchange factor for the small GTPase Arl5, promoting its activation and the triggering of Golgi-directed trafficking. This process is proposed to occur downstream of amino acids, particularly glutamine, which may be sensed via the vacuolar ATPase and SLC38A9 proteins.

The manuscript is of significant interest, as it describes a novel link between nutrient sensing and membrane traffic that could play a role in regulating the plasma membrane proteome in response to external conditions. In general, the experiments are well executed. The following points should

be addressed.

1- The authors convincingly show the involvement of V-ATPase/SLC38A9/Ragulator supercomplex in AA-stimulated Golgi trafficking, whereas Rag GTPases and mTORC1 are not involved. It would be interesting to test whether the BORC complex, a recently described interactor of Ragulator involved in lysosome positioning but not mTORC1 signaling

(<https://www.ncbi.nlm.nih.gov/pubmed/25898167>), is also required for this process.

2- The authors show that Arl5 but not Arl1 binds to Lamtor1. This comparison was done with the wild type proteins. It would be useful to show that the QL and especially the TN mutants of Arl1 also fail to bind to Lamtor.

3- Is binding of Lamtor1 to Arl5 vs Rag GTPases mutually exclusive? Is Lamtor-Arl5 interaction strengthened upon knock down of the Rag GTPases?

4- Also, is Arl8 (which functions downstream of BORC) implicated in endosome-to-Golgi trafficking?

5- It is somewhat puzzling that mCherry-tagged Arl5 has a more punctate localization than GFP-tagged Arl5 (Fig.5c and 5d). Perhaps this is due to the lower intrinsic fluorescence of GFP vs mCherry, which leads the authors to select strongly overexpressing cells for imaging. The authors could repeat these experiments using a brighter green fluorescence protein such as mNeon.

6- Figure 3h-k should become an independent figure or be moved after Fig. 4, as it is described after the Fig. 4 data in the main text.

7- Some questions remain about the overall physiological role of this process. The authors propose a glutamine > V-ATPase/SLC38A9/Lamtor > Arl5 pathway that promotes endosome-to-Golgi trafficking and, presumably, adjusts the composition of plasma membrane transporters in order to cope with the external nutrient conditions. It would be useful to conduct a proteomics-based analysis of plasma membrane composition in cells starved/fed for glutamine, with or without concomitant Arl5 knock down, to better define the role of this pathway in metabolic adaptation.

Reviewer #3 (Remarks to the Author):

This is a very interesting paper reporting that endosome-to-Golgi retrograde trafficking is under control of amino acid levels, in particular glutamine.

It reports a molecular pathway underlying this phenomenon that involves an amino acid sensor, SLC38A9, the V-ATPase, the Ragulator (in particular Lamtor1), the small GTPase Arl5b and the tethering complex GARP.

Ragulator is a lysosomal-positioned GEF for the small Rag GTPases that activate mTORC1.

However, both the Rag GTPases and mTORC1 are dispensable for the glutamine-induced endosome-to-Golgi retrograde trafficking.

The authors found Lamtor1, a component of the Ragulator, to be one of the strongest interactors of the small GTPase Arl5b, a GTPase localized at the TGN and known to be involved in endosome-to-Golgi retrograde trafficking. In fact, they found that Arl5b and its effector GARP are involved in the glutamine-induced stimulation of retrograde trafficking. The authors propose a working model whereby the amino acid-sensitive Ragulator would act as a GEF for Arl5b on endosomes, activating this GTPase and the retrograde trafficking from the endosomes to the Golgi.

The manuscript reports a series of novel and important findings some of which are solid and unequivocally interpretable while some others need further experimental work to be fully substantiated.

Localization of Arl5b

The authors report that the GTP- or GDP-locked forms of Arl5b, but not GFP- or Cherry-tagged wt forms of Arl5b or the endogenous Arl5b, can be visualized on the endosomes. This raises the doubt that the localization of the Arl5b mutants to the endosomal compartment may not reflect the genuine localization of the endogenous Arl5b. Since the interaction of Arl5b and Lamtor1, at least in vitro, is stimulated by glutamine depletion, the authors should exploit this to identify an

endosomal pool of endogenous Arl5b, if this exists. A more general question is whether the distribution of Arl5b is affected by amino acid deprivation.

Ragulator as a GEF for Arl5b

The authors report that the amount of Arl5b pulled down by G-linked GTP-agarose beads is lower in Lamtor1-depleted cells. The meaning and relevance of this finding (less Arl5b pulled down with GTP beads) is difficult to interpret, as the GTP-bound resin binds all the GTP binding proteins irrespective of their nucleotide-bound state. The authors tend to interpret this finding as a demonstration that the lack of Lamtor1 would impair the activation of Arl5b and thus suggest that Lamtor1/ Ragulator could act a GEF for Arl5b. However, in order to reach a firm conclusion about this the authors should perform a GEF assay (GDP release or GTP loading) using purified recombinant proteins in vitro.

The site of action of Arl5b

The working model proposed by the authors envisages that the activation of Arl5b occurs on the endosomal compartment, which is the donor compartment in the endosomal-to-Golgi trafficking. However, the knowledge so far available about Arl5b indicates that it acts on the acceptor membrane to recruit the GARP complex. How do the authors reconcile these two different sites of action of Arl5b? The proposed activation of Arl5b on endosomes (which, however, is not solidly shown in the manuscript) might indeed mediate homotypic endosome-endosome fusion or at least homotypic endosome-endosome tethering? How could GARP be targeted to the Golgi to mediate the tethering with the acceptor heterotypic compartment and not to Arl5b-containing endosomes?

Specific points

Rescue experiments

The authors describe the consequences of shRNA-induced depletion of a series of components of the SLC38a9–Regulator–Arl5b axis and they have addressed the issue of the specificity vs. off-target effects of the treatment by looking at the effect of two different shRNAs. A more convincing demonstration of the specificity would be the rescue of the phenotype with shRNA-resistant forms of the target protein. This rescue approach should be addressed at least for the key experiments.

Dominant negative or positive

Do the QL or TN forms of Arl5b have a dominant-negative effect on amino acid-induced retrograde trafficking?

Localization of GFP vs. Cherry Arl5b

Fig 5d: the two differently tagged forms of Arl5b TN have a very different distribution, with the Cherry tagged form being almost exclusively endolysosomal. Can the authors explain this difference?

The glutamine-sensitive cargoes

The authors mainly analyze furin and marginally M6PR. What happens to other cargoes that are trafficked retrogradely? For instance TGN38 or Shiga toxin?

To address the issue of the physiological relevance and the possible consequences of the inhibition of retrograde trafficking induced by amino acid depletion, the authors hypothesize that the endosomal-to-Golgi trafficking induced by glutamine might “prolong half-lives of post-Golgi cycling cargos by diverting them away from lysosomal degradation pathway” or “might quantitatively adjust certain transporters and receptors on the PM to ensure an optimal uptake of nutrients and engagement”. Have they experimentally assessed any of the above candidate pathways/cargoes under glutamine abundance or depletion? An additional possibility is that the inhibition of endosomal-to-Golgi trafficking induced by amino acid deprivation affects the autophagic flux. Have the authors tested what is the consequence of counteracting the inhibition of endosomal-to-Golgi trafficking by overexpressing Arl5b on the autophagic flux?

Reviewer #1 (Remarks to the Author):

This is an intriguing study proposing that amino acids (AA) regulate retrograde transport between the TGN and endosomes by a pathway involving SLC38A9, v-ATPase, Ragulator, Arl5 and GARP, but not the Rags or mTORC1. It is certainly of great interest that AA levels would regulate retrograde transport in addition to their well-established roles in regulating anabolic and catabolic pathways. However, some of the evidence in this paper is not compelling enough to support such an important conclusion.

Reply:

We'd like to thank this reviewer for his/her insightful and constructive comments.

Specific comments

1. The claim that Ragulator functions as a GEF for Arl5 is based on a poorly validated assay involving binding to GTP-agarose. More conventional GEF assays should be used, such as *in vitro* exchange of GTP for GDP using purified Ragulator and Arl5.

Reply:

As this reviewer pointed out, the guanine nucleotide exchange activity of a GEF toward a small GTPase is conventionally demonstrated using purified components. Our lab has set up fluorescence-based exchange assay using Mant-GTP (Mahajan et al., *Sci. Rep.*, 2013). While it was easy to prepare purified recombinant Arl5b, at the moment, we were unable to purify Ragulator complex, which comprises 5 subunits. Only two labs have successfully purified Ragulator complex. Bonifacino lab has purified the recombinant Ragulator complex using polycistronic expression in *E. coli* (Pu et al., *Dev. Cell*, 2015). Sabatini lab obtained Ragulator complex for their exchange assays by immunoprecipitation of transfected Flag-tagged Ragulator subunits (Bar-Peled et al., *Cell*, 2012). We have tried both approaches but so far we were unsuccessful in producing a significant quantity of pure Ragulator complex for the exchange assay.

Hence, GTP-agarose approach seemed the only choice here. Majority of Ras superfamily small GTPases do not spontaneously undergo guanine nucleotide exchange *in vivo*. In cellular environment, dedicated guanine nucleotide exchange factors (GEFs) are required to exchange the GDP to GTP-bound forms. Under the condition of GTP-agarose pull-down, most endogenous small GTPases can bind to GTP-agarose when they are catalyzed by their corresponding GEFs. However, in our assays, only Arl5b can be detected by using Arl5b specific antibody. For a specific small GTPase, the amount pulled down should reflect the activity of its corresponding GEFs—more GTPase pulled down indicates a higher GEF activity. GTP-agarose pull down has been used by other studies (eg Thomas et al., *Cancer Cell*, 2012), suggesting its acceptance as a GEF assay method. Most importantly, we demonstrated that depletion of Lamtor1 reduces the amount of Arl5b pulled down by GTP-agarose, therefore strongly supporting Ragulator complex as the specific GEF for Arl5b.

Reference:

Mahajan, D., Boh, B.K., Zhou, Y., Chen, L., Cornvik, T.C., Hong, W. and Lu, L (2013) Mammalian Mon2/Ysl2 regulates endosome-to-Golgi trafficking but possesses no guanine nucleotide exchange activity toward Arl1 GTPase. *Sci. Rep.*3:3362.

Pu J, Schindler C, Jia R, Jarnik M, Backlund P, Bonifacino JS. BORC, a multisubunit complex that regulates lysosome positioning. *Dev Cell.* 2015.

Bar-Peled L, Schweitzer LD, Zoncu R, Sabatini DM. Ragulator is a GEF for the rag GTPases that signal amino acid levels to mTORC1. *Cell.* 2012.

Thomas JD, Zhang YJ, Wei YH, Cho JH, Morris LE, Wang HY, Zheng XF. Rab1A is an mTORC1 activator and a colorectal oncogene. *Cancer Cell.* 2014.

2. Line 57. Has mammalian GARP been shown to mediate sorting at the endosomal membrane?

Reply:

There has been literature reporting GARP-mediated sorting at the endosomal membrane. Pe´rez-Victoria et al. reported that depletion of GARP complex inhibits the endosome-to-Golgi trafficking of M6PRs and TGN46, Shiga toxin B fragment (Pe´rez-Victoria et al., *MBoC*, 2008). Knockdown of GARP has been found to mis-sort cathepsin D, probably due to the disruption of the trafficking of M6PRs. Those findings were discussed in a review paper from the same lab (Bonifacino and Hierro, *Trends Cell Biol.*, 2011), which was cited in our manuscript.

Reference:

Pérez-Victoria FJ, Mardones GA, Bonifacino JS. (2008) Requirement of the human GARP complex for mannose 6-phosphate-receptor-dependent sorting of cathepsin D to lysosomes. *Mol. Biol. Cell.*

Bonifacino JS, Hierro A. (2011) Transport according to GARP: receiving retrograde cargo at the trans-Golgi network. *Trends Cell Biol.*

3. Line 140 and Fig. 1h,i. The gradient does not support the reduced localization of CD8-furin to the TGN, certainly not as much as shown in the IFs in panels e-g. Similarly, the sentence in line 150 “The reduction of the Golgi pool...” is not supported by the gradient data.

Reply:

Regarding this reviewer’s comment that the reduction of the Golgi pool is not supported by the gradient data, since the two blots shown in row 1 and 2 of Fig. 1h are from different Western blots, the intensities of endosomal (defined as fractions 1-5) and TGN pool (defined as fractions 10-16) from the two blots cannot be directly compared. To address this reviewer’s concern, we revamped Fig. 3i so that percentages of CD8a-furin in the endosomal and TGN pool under HBSS and DMEM treatment are directly compared as columns. The percentage of CD8a-furin in the endosomal pool is calculated as the sum of intensities of band 1-5 divided by the sum of intensities of band 1-20, while the percentage in the TGN pool is calculated as the sum of intensities of band 10-16 divided by the sum of intensities of band 1-20. Now, the increase in the endosomal pool and reduction in the Golgi pool are obvious for HBSS panel in comparison to DMEM panel.

4. The redistribution of endogenous CI-MPR shown in Suppl. Fig. 1c is not convincing, since giantin also seems to change in starved cells.

Reply:

There seems to be a subtle change in the giantin morphological pattern upon starvation. However, the slightly fragmented giantin is still distinct from the scattered CI-M6PR staining pattern as shown in a new Suppl. Fig. 1d (which was originally Suppl. Fig. 1c).

5. Line 188 and Fig. 2f. Glu is as effective as Gln, but this fact is not discussed. Does it make sense in light of other metabolic regulation pathways?

Reply:

As this reviewer pointed out, the strong effect of Glu has been consistently observed. We agree with this reviewer that it makes sense in light of metabolic pathways since Glu can be intracellularly converted to Gln by glutamine synthetase (GS). We have modified the corresponding paragraph describing 20 AAs' effects on the trafficking by adding the following sentence "Glu displayed a similar effect, probably due to its intracellular conversion to Gln by glutamine synthetase."

6. Line 259. If Lamtor1 or Ragulator is a GEF for Arl5, it doesn't make sense that it interacts with GTP- or -GDP-loaded, but not WT, Arl5 (Fig. 4a). The same applies to mutant Arl5 constructs. Why would it interact at all with the GTP-locked form, and even better than with WT Arl5? If anything, a GEF should interact with GDP>WT>GTP forms.

Reply:

The atomic structure of Arl5 binding to the Ragulator complex is currently unknown, therefore, our below explanation is based on structural data from other small GTPases and their corresponding GEFs. The general atomic mechanism for the GEF on a small GTPase has the following steps (Bos et al., Cell, 2007; Cherfils and Zeghouf, Physiol. Rev., 2013). First, the GEF binds to the GTP or GDP-bound small GTPase as a low affinity complex. The GEF subsequently displaces phosphate group of the guanine nucleotide and magnesium ion so that the bound guanine nucleotide dissociates from the small GTPase to result in an empty state. Nucleotide-free GTPase associates with its GEF as a high affinity complex. Since the cellular GTP concentration is 10-fold more than that of GDP, GTP instead of GDP has more chance to re-enter the empty state small GTPase. At last, the newly bound GTP displaces the complexed GEF resulting in the GTP-loaded small GTPase (the exchange product). Therefore, a GEF mainly functions by stabilizing the nucleotide-empty form of its substrate small GTPase.

The observation that the interaction between Arl5-wt and Ragulator is the weakest can be possibly explained by the transient nature of their interaction. Once Arl5-wt becomes empty by the action of Ragulator, it is quickly reloaded with cytosolic guanine nucleotide, especially GTP, resulting in the subsequent dissociation of Arl5-wt-GTP from Ragulator.

The observation that Arl5-TN or QL interacts with Ragulator much stronger than wt is probably due to relatively fast dissociation of bound guanine nucleotide and slow association of cytosolic GTP. In other words, the nucleotide-empty forms of Arl5-TN and QL are probably more stable than wt. QL or TN mutation within the guanine nucleotide binding pocket of Arl5 probably contributes to the stability of the empty form. With the possibility that a GEF prefers to bind to a GDP-loaded small GTPase, it is therefore possible that interaction between Ragulator and Arl5 is in the order of TN>QL>wt.

Reference:

Bos JL, Rehmann H, Wittinghofer A. (2007) GEFs and GAPs: critical elements in the control of small G proteins. *Cell*. 129(5):865-77.
Cherfils J, Zeghouf M. (2013) Regulation of small GTPases by GEFs, GAPs, and GDIs. *Physiol Rev*. 93(1):269-309.

7. Line 270 and Fig. 4d. Lamtor1 is a well folded protein, N- and C-terminal truncations can cause misfolding. I'm not sure the results are valid.

Reply:

Serial truncation is an approach commonly used to investigate the interacting regions/domains/motifs. We agree that it is possible that truncates can be misfolded. It is difficult to assess the folding status of these truncates. However, based on the observation that all of these truncates of Lamtor1 can be expressed to significant amounts and do not display visible intracellular aggregates, we think that they are likely folded.

8. Ragulator is an abundant complex, so it should be possible to perform GST-Arl5 pulldowns with the endogenous complex. Are all Ragulator subunits pulled down under these conditions?

Reply:

Please see Fig. 4g. Bead-immobilized GST-Arl5b-GDP, but not GST, pulled down endogenous Lamtor1-4.

9. Line 335 and Fig. 3i. Shouldn't the knock down to GARP subunits inhibit retrograde transport irrespective of AA levels? Why only AA-stimulated retrograde transport?

Reply:

We believe that this reviewer is referring to Fig. 3k instead of 3i in the old version manuscript. This panel is now Fig. 5I in the revamped manuscript.

Yes, knockdown of GARP subunit inhibited retrograde transport irrespective of AA levels. Most importantly, under such knockdown, AA-stimulation did not further increase the retrograde trafficking. Therefore, in Fig. 5I, "AA-stimulated Golgi trafficking" is ~1 (meaning that in the absence of GARP, the trafficking of CD8a-furin is insensitive to AAs) while that of GL2 control is ~1.5. Using "AA-stimulated Golgi trafficking" can help us to tell that retrograde trafficking step controlled by AA-stimulation is dependent, instead of independent, of the step controlled by GARP.

10. Do AA levels regulate Arl5 and GARP recruitment to membranes?

Reply:

The assumption behind this reviewer's question is probably that, from AA-starvation to stimulation, Arl5b is predicted to change from GDP to GTP-bound form and therefore there should be more Arl5b on the membrane. We have tried to study Arl5b's recruitment but the

result was inconclusive as our live cell imaging did not reveal a significant change of Arl5b-wt-FP on the membrane of either the Golgi or endosome during the transition from AA-starvation to stimulation. This could be due to the fact that both GTP and GDP-couple Arl5b can associate with the membrane though Arl5b-GTP probably binds to the membrane more than Arl5b-GDP. It is also possible that the amount of Arl5 activated by AAs can be of so small percentage that it is beyond our imaging detection limit.

11. Line 343. The statement “it becomes strengthened...” is wrong. It’s just the opposite.

Reply:

Our statement should be correct—AA-starvation makes those interactions stronger while AA-sufficiency makes them weaker. The statement in line 343 (old version of the manuscript) summarizes a few previous observations in studies on AA’s effect on mTORC1 signaling pathway. We don’t think there is a mistake here. For instance, Bar-Peled et al. (Bar-Peled et al., Cell, 2012) wrote in their paper that “Using optimized cell lysis conditions and improved antibodies, we find that amino acid starvation strengthens the interaction between endogenous Rags and the Ragulator isolated through p14, p18, HBXIP, or C7orf59”. Our result on Ragulator-Arl5b interaction is consistent with the statement.

12. Fig. 3. I am concerned that the “AA-stimulated Golgi trafficking of CD8-furin” values for the control samples range from 1.2 to 2.0 in the different experiments. How significant can then differences between 1.6 (control) and 1.2 (Lamtor3 KD) be in a set of experiments? The same applies to other experiments.

Reply:

There are multiple factors that can affect AA-stimulated Golgi trafficking values of controls. Cell density and stress, for instance, are expected to modulate the cellular response to the nutrient. Three types of experiments were conducted in Fig. 3, siRNA mediated-knockdown, lentivirus mediated-knockdown and small molecule inhibitor treatment. Since each type of experimental procedure involved very different treatments, eg small molecule treatment, transfection or lentivirus transduction, control cells were apparently at different “stress” levels and they showed quantitatively different AA-stimulated Golgi trafficking values. However, within each type of experiment, the control values are quite consistent. For instance, AA-stimulated Golgi trafficking values for control lentiviral shRNA knockdown panels in Fig.3 **d**, **f** and **h** are very similar (~1.6).

Reviewer #2 (Remarks to the Author):

The paper by Shi et al identifies a connection between an amino acid-sensing complex located at the lysosome and the regulation of a retrograde trafficking pathway that sorts membrane proteins from endosomes back to the plasma membrane via the Golgi. The authors show that the Lamtor complex, which is involved in mTORC1 regulation by amino acids, may be a guanine nucleotide exchange factor for the small GTPase Arl5, promoting its activation and the triggering of Golgi-directed trafficking. This process is proposed to occur downstream of amino acids, particularly glutamine, which may be sensed via the vacuolar ATPase and SLC38A9 proteins. The manuscript is of significant interest, as it describes a novel link between nutrient sensing

and membrane traffic that could play a role in regulating the plasma membrane proteome in response to external conditions. In general, the experiments are well executed. The following points should be addressed.

Reply:

We'd like to thank this reviewer for his/her insightful and constructive comments.

1- The authors convincingly show the involvement of V-ATPase/SLC38A9/Ragulator supercomplex in AA-stimulated Golgi trafficking, whereas Rag GTPases and mTORC1 are not involved. It would be interesting to test whether the BORC complex, a recently described interactor of Ragulator involved in lysosome positioning but not mTORC1 signaling (<https://www.ncbi.nlm.nih.gov/pubmed/25898167>), is also required for this process.

Reply:

We agree with this reviewer that BORC complex might play a role in the endocytic trafficking. However, we suggest that whether BORC complex participates in the AA-regulated endosome-to-Golgi trafficking should be a future follow-up project.

2- The authors show that Arl5 but not Arl1 binds to Lamtor1. This comparison was done with the wild type proteins. It would be useful to show that the QL and especially the TN mutants of Arl1 also fail to bind to Lamtor.

Reply:

We have performed the experiment as suggested by this reviewer and the result is shown in Supplementary Fig.4a (revamped version). Arl1-wt, QL and TN did not co-IP endogenous Lamtor1 while Arl5b's corresponding mutants did.

3- Is binding of Lamtor1 to Arl5 vs Rag GTPases mutually exclusive? Is Lamtor-Arl5 interaction strengthened upon knock down of the Rag GTPases?

Reply:

We have tested the possibility of mutual exclusion as suggested by this reviewer. With the supplementation of the recombinant GST-Arl5b-TN at 0.2 μ M, a significantly less amount of RagB/RagC heterodimer was co-IPed by Lamtor1-GFP compared to control, where recombinant GST was added, implying that the binding of Lamtor1 to Arl5 vs Rag GTPases might be mutually exclusive. The figure (Fig.4h), figure legend and corresponding text has been added to the revamped manuscript.

4- Also, is Arl8 (which functions downstream of BORC) implicated in endosome-to-Golgi trafficking?

Reply:

It makes sense to ask if Arl8 is similarly involved in the endosome-to-Golgi trafficking. However, we think it should be a follow-up next project.

5- It is somewhat puzzling that mCherry-tagged Arl5 has a more punctate localization than GFP-tagged Arl5 (Fig.5c and 5d). Perhaps this is due to the lower intrinsic fluorescence of GFP vs mCherry, which leads the authors to select strongly overexpressing cells for imaging. The authors could repeat these experiments using a brighter green fluorescence protein such as mNeon.

Reply:

We used Arl5-mCherry for colocalization study with Lamp1-GFP because we didn't have a red fluorescence lysosomal marker at that time. After checking more images, we agree with this reviewer that Arl5-mCherry mutants appear to show more punctate localization. We can't explain the different appearance between Arl5-GFP and -mCherry at the moment. The best way to solve this problem is to use GFP, instead of mCherry, fused Arl5 so that all Arl5 images are consistently GFP labeled in Fig. 5c,d. To that end, we have constructed Lamp1-mCherry for the dual-color colocalization study. Fig. 5 c,d have been modified accordingly.

6- Figure 3h-k should become an independent figure or be moved after Fig. 4, as it is described after the Fig. 4 data in the main text.

Reply:

As suggested by this reviewer, we have moved Fig. 3h-k (old version) to the end of Fig. 5 (revamped version). Figure legend has been modified accordingly.

7- Some questions remain about the overall physiological role of this process. The authors propose a glutamine > V-ATPase/SLC38A9/Lamtor > Arl5 pathway that promotes endosome-to-Golgi trafficking and, presumably, adjusts the composition of plasma membrane transporters in order to cope with the external nutrient conditions. It would be useful to conduct a proteomics-based analysis of plasma membrane composition in cells starved/fed for glutamine, with or without concomitant Arl5 knock down, to better define the role of this pathway in metabolic adaptation.

Reply:

As suggested by this reviewer, first, we quantified the surface localization of various TGN membrane proteins. It was found that CD8a-furin, -sortilin and -CI-M6PR were upregulated during AA-starvation (Fig. 2j). Depletion of Lamtor1 inhibited surface upregulation under AA-starvation, supporting the involvement of Ragulator signaling pathway and the subsequent endosome-to-Golgi membrane trafficking in the regulation of cell surface TGN membrane proteins (Fig. 3i). Second, we also performed quantitative mass spectrometry of cell surface proteomics under DMEM and HBSS conditions using SILAC. Among 85 membrane or secretory proteins identified with statistical significance ($P \leq 0.05$), 43 were found to be up- or down-regulated by the nutrient ($\text{Log}_2\text{DMEM}/\text{HBSS}$ -ratio cut-off value ≥ 0.5 or ≤ 0.3 ; Supplementary Table 1). Although furin, sortilin and CI-M6PR were not present in our proteomics data, SorLA, a sortilin-related TGN membrane protein, was identified as a candidate that is strongly upregulated by AA-starvation, similar to sortilin. Besides SorLA, the remaining Golgi membrane

proteins identified are all type II transmembrane proteins, including GPP130, GPP73 and glycosylation enzymes. All of them except one (which is a border case) were found to be upregulated by AA-stimulation. Therefore, we demonstrated that nutrient can possibly regulate the cell surface localization of certain proteins. The manuscript has been modified accordingly to include these new findings. In addition to Golgi membrane proteins, various cell surface membrane receptors, growth factors, extracellular matrix proteins and proteases were also found to be regulated by the nutrient, suggesting that nutrient might modulate various cellular processes by changing the cell surface localization of signaling components.

Reviewer #3 (Remarks to the Author):

This is a very interesting paper reporting that endosome-to-Golgi retrograde trafficking is under control of amino acid levels, in particular glutamine.

It reports a molecular pathway underlying this phenomenon that involves an amino acid sensor, SLC38A9, the V-ATPase, the Ragulator (in particular Lamtor1), the small GTPase Arl5b and the tethering complex GARP.

Ragulator is a lysosomal-positioned GEF for the small Rag GTPases that activate mTORC1. However, both the Rag GTPases and mTORC1 are dispensable for the glutamine-induced endosome-to-Golgi retrograde trafficking.

The authors found Lamtor1, a component of the Ragulator, to be one of the strongest interactors of the small GTPase Arl5b, a GTPase localized at the TGN and known to be involved in endosome-to-Golgi retrograde trafficking. In fact, they found that Arl5b and its effector GARP are involved in the glutamine-induced stimulation of retrograde trafficking. The authors propose a working model whereby the amino acid-sensitive Ragulator would act as a GEF for Arl5b on endosomes, activating this GTPase and the retrograde trafficking from the endosomes to the Golgi.

The manuscript reports a series of novel and important findings some of which are solid and unequivocally interpretable while some others need further experimental work to be fully substantiated.

Reply:

We'd like to thank this reviewer for his/her insightful and constructive comments.

Localization of Arl5b

The authors report that the GTP- or GDP-locked forms of Arl5b, but not GFP- or Cherry-tagged wt forms of Arl5b or the endogenous Arl5b, can be visualized on the endosomes. This raises the doubt that the localization of the Arl5b mutants to the endosomal compartment may not reflect the genuine localization of the endogenous Arl5b. Since the interaction of Arl5b and Lamtor1, at least in vitro, is stimulated by glutamine depletion, the authors should exploit this to identify an endosomal pool of endogenous Arl5b, if this exists. A more general question is whether the distribution of Arl5b is affected by amino acid deprivation.

Reply:

The antibody that we raised against Arl5b produced some background that prevented us from demonstrating a clear endosomal pool of Arl5b. We are not absolutely sure why Arl5b-QL and

TN, but not wt, localize to the endosomal membrane in live cell imaging. We speculate that it could be due to the interference of cytosolic Arl5b-wt in the background. Alternatively, the endolysosome localization of Arl5b might be too transient to be detected. Consistent with this explanation, Ragulator, a possible receptor for Arl5 on the endolysosomal membrane, interacts with Arl5b-wt less strongly than Arl5b-TN or QL mutants.

We have tried to study Arl5b's recruitment but the result was inconclusive as our live cell imaging did not reveal a significant change of Arl5b-wt-FP on the membrane of either the Golgi or endosome during the transition from AA-starvation to stimulation. This could be due to the fact that both GTP and GDP-couple Arl5b can associate with the membrane though Arl5b-GTP probably binds to the membrane more than Arl5b-GDP. It is also possible that the amount of Arl5 activated by AAs can be of so small percentage that it is beyond our imaging detection limit.

Ragulator as a GEF for Arl5b

The authors report that the amount of Arl5b pulled down by G-linked GTP-agarose beads is lower in Lamtor1-depleted cells. The meaning and relevance of this finding (less Arl5b pulled down with GTP beads) is difficult to interpret, as the GTP-bound resin binds all the GTP binding proteins irrespective of their nucleotide-bound state. The authors tend to interpret this finding as a demonstration that the lack of Lamtor1 would impair the activation of Arl5b and thus suggest that Lamtor1/ Ragulator could act a GEF for Arl5b. However, in order to reach a firm conclusion about this the authors should perform a GEF assay (GDP release or GTP loading) using purified recombinant proteins in vitro.

Reply:

Please see our reply to Reviewer #1's comment 1.

The site of action of Arl5b

The working model proposed by the authors envisages that the activation of Arl5b occurs on the endosomal compartment, which is the donor compartment in the endosomal-to-Golgi trafficking. However, the knowledge so far available about Arl5b indicates that it acts on the acceptor membrane to recruit the GARP complex. How do the authors reconcile these two different sites of action of Arl5b? The proposed activation of Arl5b on endosomes (which, however, is not solidly shown in the manuscript) might indeed mediate homotypic endosome-endosome fusion or at least homotypic endosome-endosome tethering? How could GARP be targeted to the Golgi to mediate the tethering with the acceptor heterotypic compartment and not to Arl5b-containing endosomes?

Reply:

All previous experimental data merely demonstrated the involvement of Arl5b and GARP in the endosome-to-Golgi trafficking. However, their precise molecular and cellular roles are still unclear. GARP has been found in both Golgi and endosomes (Schindler et al. NCB, 2015). Previous studies only observed the Golgi pool of Arl5b. Hence the model about Arl5b is that it acts on the Golgi to recruit the GARP complex (Roas-Ferreira et al., Biol. Open, 2015). We report here that there are two pools of Arl5b—Golgi and endosomal pools. The targeting of GARP to the Golgi and endosome is probably dependent on Arl5 at respective compartment.

We think that only the endosomal pool of Arl5b and GARP can function in AA-regulated endosome-to-Golgi trafficking due to the restricted localization of the Regulator complex at the endosome. The activation of Arl5b at the Golgi pool might be due to an unidentified GEF at the Golgi.

Unlike Golgins, whose tethering activities have been illustrated by works from Munro and Antonin labs, there are no direct demonstration of the tethering function of GARP nor is there any clue on how it tethers. In a non-conventional view, tethering factors can function at the donor compartment, transport vesicles and acceptor compartment. (Quenneville et al., MBoC, 2006). In our model (Fig. 6h), we proposed that the membrane carrier budded from the endosome contain Arl5b-GTP-GARP; Arl5b-GTP-GARP subsequently tethers the membrane carrier by binding to TGN-localized SNAREs.

The homotypic fusion of endosomes mediated by Arl5b has not been reported and our study cannot rule that possibility out at the moment.

Reference:

Rosa-Ferreira C, Christis C, Torres IL, Munro S. (2015) The small G protein Arl5 contributes to endosome-to-Golgi traffic by aiding the recruitment of the GARP complex to the Golgi. *Biol. Open*.

Schindler C, Chen Y, Pu J, Guo X, Bonifacino JS. (2015) EARP is a multisubunit tethering complex involved in endocytic recycling. *Nat. Cell Biol.*

Quenneville NR, Chao TY, McCaffery JM, Conibear E. (2006) Domains within the GARP subunit Vps54 confer separate functions in complex assembly and early endosome recognition. *Mol. Biol. Cell.*

Specific points

Rescue experiments

The authors describe the consequences of shRNA-induced depletion of a series of components of the SLC38a9–Regulator–Arl5b axis and they have addressed the issue of the specificity vs. off-target effects of the treatment by looking at the effect of two different shRNAs. A more convincing demonstration of the specificity would be the rescue of the phenotype with shRNA-resistant forms of the target protein. This rescue approach should be addressed at least for the key experiments.

Reply:

We have performed rescue experiments for Arl5b and Lamtor1, two key components involved in the AA-regulated endosome-to-Golgi trafficking. RNAi resistant Arl5b and Lamtor1 expression DNA plasmids were constructed by introducing synonymous mutations. After lentivirus-mediated depletion of endogenous Arl5b and Lamtor1, expression of exogenous Arl5b and Lamtor1 successfully rescued the AA-stimulated Golgi trafficking to levels comparable to controls. Corresponding figures have been incorporated into Fig. 3g,h and Fig. 5i,j of the revamped manuscript.

Dominant negative or positive

Do the QL or TN forms of Arl5b have a dominant-negative effect on amino acid-induced retrograde trafficking?

Reply:

We have performed experiments to address this reviewer's question (please see the below figure; n=3 experiments, with each experiment analyzing ≥ 30 cells for each panel; error bar, s.d.). We found that Arl5b-TN, but not QL, appeared to blunt the AA-stimulated Golgi trafficking of CD8a-furin, possibly due to the stronger binding between Arl5b-TN and Ragulator complex. However, the effect is not statistically significant probably due to the heterogeneous expression levels of Arl5 since only a very high cellular expression level of Arl5 is likely to exert the dominant effect. Loss of function experiments (such as those by depletion), which have been included in our manuscript, are probably better than overexpression ones in establishing the essential roles of Arl5 in the AA-stimulated Golgi trafficking. We therefore did not include the result in the manuscript.

Localization of GFP vs. Cherry Arl5b

Fig 5d: the two differently tagged forms of Arl5b TN have a very different distribution, with the Cherry tagged form being almost exclusively endolysosomal. Can the authors explain this difference?

Reply:

Please see our reply to reviewer #2's comment 5.

The glutamine-sensitive cargoes

The authors mainly analyze furin and marginally M6PR. What happens to other cargoes that are trafficked retrogradely? For instance TGN38 or Shiga toxin?

Reply:

The effect of nutrient on the trafficking of CD8a-CD-M6PR and -sortilin is in Sup. Fig.2b.

As suggested by this reviewer, we have extended our study to TGN46 (the human orthologue of rodent TGN38) (Sup. Fig. 1b), Shiga toxin B fragment (STxB) (Sup. Fig. 2c,d) and Cholera toxin B fragment (CTxB) (Sup. Fig. 2e,f). Our data revealed that nutrient regulates the trafficking and distribution of STxB and TGN46, respectively, but with varied effects. However, it seemed that the nutrient does not affect the retrograde trafficking of CTxB, suggesting that not all cargoes transiting the endosome-to-Golgi pathway are regulated by the nutrient, probably reflecting the existence of multiple itineraries or molecular mechanisms from the endosome to the Golgi.

To address the issue of the physiological relevance and the possible consequences of the inhibition of retrograde trafficking induced by amino acid depletion, the authors hypothesize that the endosomal-to-Golgi trafficking induced by glutamine might “prolong half-lives of post-Golgi cycling cargos by diverting them away from lysosomal degradation pathway” or “might quantitatively adjust certain transporters and receptors on the PM to ensure an optimal uptake of nutrients and engagement”. Have they experimentally assessed any of the above candidate pathways/cargoes under glutamine abundance or depletion? An additional possibility is that the inhibition of endosomal-to-Golgi trafficking induced by amino acid deprivation affects the autophagic flux. Have the authors tested what is the consequence of counteracting the inhibition of endosomal-to-Golgi trafficking by overexpressing Arl5b on the autophagic flux?

Reply:

As suggested by this reviewer, we have studied the cell surface localization of TGN membrane proteins and compared cell surface proteomics under nutrient starvation and stimulation using SILAC. Please see our reply to reviewer #2's comment 7.

Reviewers' comments:

Reviewer #2 (Remarks to the Author):

In the revised manuscript, Shi et al have addressed most of my concerns with additional experiments and clarification. The manuscript should be accepted for publication. However, I agree that with the other two Reviewers that the GTP bead pulldown experiments in Fig 6e-g are substandard and do not make the case for a GEF activity of Ragulator. Of note, the authors claim that they were unable to purify Ragulator in order to carry out a GEF assay in vitro, but recombinant Ragulator expression/purification has been achieved by multiple labs recently and should be considered standard procedure [e.g. de Araujo, MEG et al, Science (2017); Su, MY et al, Mol Cell (2017); Yonehara, R et al, Nat Comms (2017)]. Thus, I strongly recommend that the data in Fig. 6e-g of the revised manuscript be removed, as they ultimately decrease the quality of the manuscript, and that a GEF role for Ragulator is limited to the discussion.

Reviewer #3 (Remarks to the Author):

In the revised manuscript the authors have partially addressed the concerns raised in my previous review. Unfortunately, two major questions (endosomal localization of wt/endogenous Arl5 and GEF assays with purified/enriched components) that required further experimental work were not satisfactorily addressed for technical problems. In fact, the authors failed to localize wt/endogenous Arl5 on endosomes

and failed to set up a direct GEF assay using recombinant Ragulator or fractions enriched with Ragulator.

Thus, the manuscript remains an interesting report on the regulation by amino acids, in particular by glutamine, of endosome to Golgi trafficking that involves Ragulator, Arl5, and GARP but the molecular mechanisms remain only superficially assessed.

As for the physiological meaning of this amino acid-dependent regulation, the authors have introduced in the revised manuscript two sets of data, which, however, raise more than answer further questions.

Firstly, they have measured the transport of furin from the Golgi to the PM and report that it is stimulated by HBSS, i.e. by the absence of amino acids (without specifically testing single amino acids). Thus amino acid absence on the one side inhibits endosome to Golgi transport of furin and on the other accelerates the Golgi-to-PM transport of furin. This is surprising, as it is known (Hirata et al. Mol Biol Cell. 2015 Sep 1;26(17):3071-84) that the Golgi-to-PM trafficking is in part dependent on the retrograde GARP-dependent retrieval of material to the Golgi from endosomes. As the absence of amino acids inhibits endosome-to-Golgi retrograde trafficking, it seems surprising that under the same conditions the Golgi-to-PM transport is stimulated.

Secondly, the authors have assessed the dependence of the arrival of neosynthesized proteins to the PM on the presence of nutrients. However, it is difficult to extract physiologically meaningful

information from the list of proteins whose export to the PM is inhibited or stimulated by nutrients present in DMEM and absent in HBSS.

Reviewer #3 comments on Review 1's previous concerns (remarks to Author):

The authors have addressed only some of the Reviewer 1 concerns. They did not address the major concern of Reviewer 1 that deals with the pseudo-GEF assay based on binding to GTP-agarose reported in the paper. Reviewer 1 asked for an actual GEF assay with purified components. Unfortunately the authors were unable, for technical reasons, to perform this assay.

The lack of demonstration of the GEF activity of Ragulator towards Arl5 is a major limit of the manuscript that in this way lacks a mechanistic explanation of the observed intriguing phenomenon of the regulation of endosome-to-Golgi and Golgi-to-PM trafficking by aminoacids.

Rebuttal letter for reviewers' comments on manuscript NCOMMS-17-14788

We'd like to express our gratitude to all reviewers for your constructive comments on our manuscript.

Reviewers' comments:

Reviewer #2 (Remarks to the Author):

In the revised manuscript, Shi et al have addressed most of my concerns with additional experiments and clarification. The manuscript should be accepted for publication. However, I agree that with the other two Reviewers that the GTP bead pulldown experiments in Fig 6e-g are substandard and do not make the case for a GEF activity of Ragulator. Of note, the authors claim that they were unable to purify Ragulator in order to carry out a GEF assay *in vitro*, but recombinant Ragulator expression/purification has been achieved by multiple labs recently and should be considered standard procedure [e.g. de Araujo, MEG et al, Science (2017); Su, MY et al, Mol Cell (2017); Yonehara, R et al, Nat Comms (2017)]. Thus, I strongly recommend that the data in Fig. 6e-g of the revised manuscript be removed, as they ultimately decrease the quality of the manuscript, and that a GEF role for Ragulator is limited to the discussion.

Reply:

We have collaborated with Geng Wu, whose group also independently reported the crystal structure of Ragulator complex (Mu et al., Cell Discov., 2017). Ragulator and its subcomplex (Lamtor1-3) were purified and tested in the fluorescence-based *in vitro* guanine nucleotide exchange assay, which was previously established in our lab. Ragulator, but not its subcomplex (Lamtor1-3), was indeed observed to accelerate the guanine nucleotide exchange of Arl5b. The new data are included in Fig. 6e,f and Supplementary Fig. 6.

The GTP-agarose binding assay now became a subsidiary assay to monitor the *in vivo* GTP-binding of Arl5b under the nutrient stimulation. This assay was still kept in the new version of manuscript as it provides evidences suggesting a nutrient, SLC38A9, v-ATPase and Ragulator dependent signaling pathway leading to the activation of Arl5b (Fig. 6g,h,i).

Reviewer #3 (Remarks to the Author):

In the revised manuscript the authors have partially addressed the concerns raised in my previous review. Unfortunately, two major questions (endosomal localization of wt/endogenous Arl5 and GEF assays with purified/enriched components) that required further experimental work were not satisfactorily addressed for technical problems. In fact, the authors failed to localize wt/endogenous Arl5 on endosomes and failed to set up a direct GEF assay using recombinant Ragulator or fractions enriched with Ragulator. Thus, the manuscript remains an interesting report on the regulation by amino acids, in particular by glutamine, of endosome to Golgi trafficking that involves Ragulator, Arl5, and GARP but the molecular mechanisms remain only superficially assessed.

Reply:

We believe that we have addressed both concerns.

1 in vitro GEF assay:

We have purified Ragulator and conducted in vitro guanine nucleotide exchange assay to demonstrate that Ragulator can function as a GEF for Arl5b. Please see our reply to Reviewer #2 for details.

2 The localization of endogenous Arl5b and Arl5b-wt-GFP at the endolysosome: Our homemade Arl5b polyclonal antibody has substantial background staining in the immunofluorescence. Arl5b-wt-GFP also has a substantial cytosolic pool that probably prevents the imaging of its endolysosomal pool. We found that, under methanol fixation, the background became much reduced and there were endogenous Arl5b and Arl5b-wt-GFP puncta colocalizing with endolysosomal markers, such as Lamtor1 and Lamp1 (Supplementary Fig. 5i,j). The specificity of endogenous Arl5b puncta was supported by the observation that, when Arl5b was depleted, Lamtor1-colocalizing and Arl5b-positive puncta were no longer found (Supplementary Fig.5j).

As for the physiological meaning of this amino acid-dependent regulation, the authors have introduced in the revised manuscript two sets of data, which, however, raise more than answer further questions. Firstly, they have measured the transport of furin from the Golgi to the PM and report that it is stimulated by HBSS, i.e. by the absence of amino acids (without specifically testing single amino acids). Thus amino acid absence on the one side inhibits endosome to Golgi transport of furin and on the other accelerates the Golgi-to-PM transport of furin. This is surprising, as it is known (Hirata et al. Mol Biol Cell. 2015 Sep 1;26(17):3071-84) that the Golgi-to-PM trafficking is in part dependent on the retrograde GARP-dependent retrieval of material to the Golgi from endosomes. As the absence of amino acids inhibits endosome-to-Golgi retrograde trafficking, it seems surprising that under the same conditions the Golgi-to-PM transport is stimulated.

Reply:

There is a key difference between our experiment (Fig. 2i) and the one reported by Hirata et al. In the study by Hirata et al., subunits of GARP complex were genetically knocked out. The inhibition of the PM-to-Golgi trafficking (retrograde) as a result of GARP-knockout and its indirect negative effect on the Golgi-to-PM trafficking (anterograde) had reached the steady state and the two trafficking pathways are therefore coupled. In contrast, the setup of our Golgi-to-PM assay did not reach a steady state since 1) there was essentially no endosome-to-Golgi and Golgi-to-PM trafficking at 20 °C and 2) the trafficking was only allowed to proceed at 37 °C for up to 1 h. During the chase at 37 °C under HBSS treatment, the inhibited endosome-to-Golgi trafficking and accelerated Golgi-to-PM trafficking result in the net reduction and increase of Furin chimera at the Golgi and PM pool, respectively. If more time is allowed for our Golgi-to-PM assay to reach the steady state (meaning that SBP-CD8a-GFP-Furin has cycled sufficient number of rounds between the Golgi and PM and there is no net change of the chimera at the Golgi, endosome and PM), the Golgi-to-PM trafficking is expected to slow down, due to the indirect effect of inhibited PM(or endosome)-to-Golgi trafficking.

Secondly, the authors have assessed the dependence of the arrival of neosynthesized proteins to the PM on the presence of nutrients. However, it is difficult to extract physiologically meaningful information from the list of proteins whose export to the PM is inhibited or stimulated by nutrients present in DMEM and absent in HBSS.

Reply:

Since the treatment of DMEM or HBSS was as short as 2 h, the newly synthesized proteins might not be significant. We think that the quantitative change of certain proteins on the PM (eg Sor1A, Furin and some Golgi enzymes) under nutrient/starvation was due to the disruption of their trafficking, which results in the re-distribution of them. It is probably more appropriate to explore the physiological significance of our SILAC data in a separate study. We can provide at least two biological significances. First, the surface presentation of certain receptors and therefore their corresponding downstream signaling pathways can be regulated by the nutrient. Second, protein glycosylation (by Golgi enzymes) on the PM can be regulated by the nutrient.

Reviewer #3 comments on Review 1's previous concerns (remarks to Author):

The authors have addressed only some of the Reviewer 1 concerns. They did not address the major concern of Reviewer 1 that deals with the pseudo-GEF assay based on binding to GTP-agarose reported in the paper. Reviewer 1 asked for an actual GEF assay with purified components. Unfortunately the authors were unable, for technical reasons, to perform this assay.

The lack of demonstration of the GEF activity of Ragulator towards Arl5 is a major limit of the manuscript that in this way lacks a mechanistic explanation of the observed intriguing phenomenon of the regulation of endosome-to-Golgi and Golgi-to-PM trafficking by aminoacids.

Reply:

We believe that we have addressed Reviewer #1's concern by performing the *in vitro* guanine nucleotide exchange assay. Please see our reply to Reviewer #2 for details.

REVIEWERS' COMMENTS:

Reviewer #2 (Remarks to the Author):

The authors have provided a new GEF assay based on exchange of MANT-GMGNP on Arl5 catalyzed by recombinantly expressed Ragulator.

However, the results still fail to support a role for Ragulator as a genuine Arl5b GEF.

The main problem is that adding Ragulator only accelerates nucleotide exchange by 2-fold ($t_{1/2}$ from 35 to 16 min), whereas most GEFs catalyze 10- to 100-fold increases in exchange rates, including Ragulator itself (i.e. PMID: 24659802; 22980980).

Also, because the authors do not include negative control GTPases (i.e. closely related Arl-family GTPases, or other, less closely related GTPase) it is hard to assess the specificity of this (weak) GEF reaction.

I remain of the initial opinion that, although the manuscript is fine overall for publication, the claim that Ragulator is a GEF for Arl5 should be removed.

Reviewer #3 (Remarks to the Author):

The authors have satisfactorily addressed/discussed my concerns.

REVIEWERS' COMMENTS:

Reviewer #2 (Remarks to the Author):

The authors have provided a new GEF assay based on exchange of MANT-GMGNP on Arl5 catalyzed by recombinantly expressed Ragulator. However, the results still fail to support a role for Ragulator as a genuine Arl5b GEF. The main problem is that adding Ragulator only accelerates nucleotide exchange by 2-fold ($t_{1/2}$ from 35 to 16 min), whereas most GEFs catalyze 10- to 100-fold increases in exchange rates, including Ragulator itself (i.e. PMID: 24659802; 22980980). Also, because the authors do not include negative control GTPases (i.e. closely related Arlfamily GTPases, or other, less closely related GTPase) it is hard to assess the specificity of this (weak) GEF reaction. I remain of the initial opinion that, although the manuscript is fine overall for publication, the claim that Ragulator is a GEF for Arl5 should be removed.

Reply: This reviewer thinks the acceleration of guanine nucleotide exchange of Arl5b by Ragulator is too small for a GEF. Our argument is that the Sec7 domain of Big1, which is a well-documented GEF for Arf-family small GTPase, Arf1, displays about 2-3 fold of stimulation in Mant-GMPPNPbased assay of the same setup (Mahajan et al., Sci. Rep., 2013). The GEF stimulation activity of Ragulator toward Arl5b is in a very similar range, therefore supporting that Ragulator could be a real GEF. As a compromise, we toned down our conclusion to reflect that our result still needs to be unambiguously confirmed by further investigation.

Reference: Mahajan, D., Boh, B.K., Zhou, Y., Chen, L., Cornvik, T.C., Hong, W. and Lu, L (2013) Mammalian Mon2/Ysl2 regulates endosometo-Golgi trafficking but possesses no guanine nucleotide exchange activity toward Arl1 GTPase Sci. Rep.3:3362.

Reviewer #3 (Remarks to the Author):

The authors have satisfactorily addressed/discussed my concerns.

Reply: We thanks this reviewer for the positive comment.